



# Impact of biogenic very short-lived bromine on the Antarctic ozone hole during the 21st century

Rafael P. Fernandez[1,2], Douglas E. Kinnison[3], Jean-Francois Lamarque[3], Simone Tilmes[3] and Alfonso Saiz-Lopez[1]

[1]Department of Atmospheric Chemistry and Climate, Institute of Physical Chemistry Rocasolano, CSIC, Madrid, 28006, Spain.
[2]National Research Council (CONICET), FCEN-UNCuyo, UNT-FRM, Mendoza, 5500, Argentina.
[3]Atmospheric Chemistry, Observations & Modelling Laboratory, National Center for Atmospheric Research, Boulder, CO 80301, USA.

*Correspondence to*: Alfonso Saiz-Lopez (a.saiz@csic.es)

**Abstract.** Active bromine released from the photochemical decomposition of biogenic very short-lived bromocarbons (VSL$^{Br}$) enhances stratospheric ozone depletion. Based on a dual set of 1960-2100 coupled chemistry-climate simulations (i.e. with and without VSL$^{Br}$), we show that the maximum Antarctic ozone hole depletion increases by up to 14% when natural VSL$^{Br}$ are considered, in better agreement with ozone observations. The impact of the additional 5 pptv VSL$^{Br}$ on

Antarctic ozone is most evident in the periphery of the ozone hole, producing an expansion of the ozone hole area of ~5 million km$^2$, which is equivalent in magnitude to the recently estimated Antarctic ozone healing due to the implementation of the Montreal Protocol. We find that the inclusion of VSL$^{Br}$ in CAM-Chem does not introduce a significant delay of the modelled ozone return date to 1980 October levels, but instead affect the depth and duration of the simulated ozone hole. Our analysis further shows that total bromine-catalysed ozone destruction in the lower stratosphere surpasses that of chlorine

by year 2070, and indicates that natural VSL$^{Br}$ chemistry would dominate Antarctic ozone seasonality before the end of the 21st century. This work suggests a large influence of biogenic bromine on the future Antarctic ozone layer.

## 1 Introduction

The detection of the springtime Antarctic ozone hole (Farman et al., 1985) has been one of the great geophysical discoveries of the 20th century. The unambiguous scientific reports describing the active role of halogen atoms (i.e. chlorine and

bromine), released from anthropogenic chlorofluorocarbons (CFCs) and halons, in depleting stratospheric ozone (Molina and Rowland, 1974; McElroy et al., 1986; Daniel et al., 1999) led to the rapid and efficient implementation of the Montreal protocol in 1989 (Solomon, 1999). Since then, the consequent turnover on the anthropogenic emissions of long-lived chlorine (LL$^{Cl}$) and bromine (LL$^{Br}$) sources (Carpenter et al., 2014) has controlled the evolution of the strong springtime ozone depletion within the Antarctic vortex, and the first signs of recovery of the ozone hole became evident at the

beginning of the 21st century (WMO, 2014; Chipperfield et al., 2015; Solomon et al., 2016).





Several coordinated initiatives have been conducted by the scientific community to predict the future evolution of the stratospheric ozone layer and its impact on climate change (Eyring et al., 2007, 2010b; Austin et al., 2010; WMO, 2014). The multi-model CCMVal-2 ozone assessment (Eyring et al., 2010a) determined that even when Antarctic ozone return date to 1980 values is expected to occur around years 2045−2060, the impact of halogenated ozone depleting substances (ODS,

such as $LL^{Cl}$ and $LL^{Br}$) on stratospheric ozone photochemistry will persist until the end of $21^{st}$ century. Many studies show that dynamical and chemical processes affect the size, strength and depth of the ozone hole formation (see Solomon et al., (2015) and references therein). Ongoing research within the Chemistry-Climate Model Initiative (CCMI) (Eyring et al., 2013; Hegglin et al., 2014) includes model experiments that consider, along with the dominant $LL^{Cl}$ and $LL^{Br}$ anthropogenic emissions, an additional contribution from biogenic very short-lived bromocarbons ($VSL^{Br}$). This additional input of

bromine is required to reconcile current stratospheric bromine trends (Salawitch et al., 2010; WMO, 2014).

$VSL^{Br}$ are naturally released from biologically productive waters mainly within the tropical oceans (Warwick et al., 2006; Butler et al., 2007; Kerkweg et al., 2008), where strong convective uplifts efficiently entrain near surface air into the upper troposphere and lower stratosphere (Aschmann and Sinnhuber, 2013; Liang et al., 2014; Saiz-lopez and Fernandez, 2016). The current contribution of $VSL^{Br}$ to total stratospheric inorganic bromine is estimated to be in the range of 3−8 pptv

(Montzka et al., 2011; Carpenter et al., 2014; Navarro et al., 2015; Hossaini et al., 2016). The most accepted value for stratospheric injection is $VSL^{Br} \approx 5$ pptv, which currently represents approximately 30% of the total contribution from $LL^{Br}$ substances arising from both anthropogenic and natural origins (~7.8 pptv Halons + ~7.2 pptv $CH_3Br \approx 15$-16 pptv $LL^{Br}$). The additional stratospheric contribution of biogenic $VSL^{Br}$ improves the model/observations agreement with respect to stratospheric ozone trends between 1980 and present time (Sinnhuber et al., 2009), with strongest ozone depleting impacts

during periods of high aerosol loading within mid-latitudes (Feng et al., 2007; Sinnhuber and Meul, 2015). Although we still lack a scientific consensus with respect to the future evolution of $VSL^{Br}$ ocean source strength and stratospheric injection (Carpenter et al., 2014), it will probably increase in the future following the increase on sea surface temperature (SST) and oceanic nutrient supply, as well as due to the enhancement of the troposphere-to-stratosphere exchange (Hossaini et al., 2012; Leedham et al., 2013).

Previous chemistry-climate modelling studies considering $VSL^{Br}$ chemistry have mainly focused on improving the model vs. observed ozone trends at mid-latitudes with respect to equivalent setups considering only the dominant anthropogenic $LL^{Cl}$ and $LL^{Br}$ sources (Feng et al., 2007; Sinnhuber et al., 2009). However, those previous studies lack an in-depth timeline analysis of the $VSL^{Br}$ impact on the ozone hole evolution during the current century. More recently, Oman et al., (2016) determined that the addition of 5 pptv $VSL^{Br}$ to the stratosphere could delay the ozone return date to 1980 levels

by as much as one decade. Their result is in agreement with that of Yang et al., (2014), who performed present-day timeslice simulations to address the sensitivity of stratospheric ozone to a speculative doubling of $VSL^{Br}$ sources under different $LL^{Cl}$ scenarios. Even when those works addressed the important question of the return date, conclusions were obtained considering a unique simulation member for each case and an approximate approach of $VSL^{Br}$ ocean emissions. Here, using



the CAM-Chem model (Saiz-Lopez et al., 2012; Fernandez et al., 2014; Tilmes et al., 2015, 2016), we present a coherent ensemble of coupled (with an interactive ocean) chemistry-climate simulations from 1960 to 2100 with and without the contribution of oceanic $VSL^{Br}$ sources. We focus on natural $VSL^{Br}$-driven changes in the chemical composition and evolution of the Antarctic ozone hole during the 21$^{st}$ century, particularly on their influence on the seasonality and

5 enlargement of the ozone hole area, ozone hole depth and return date to 1980s levels. The analysis shown here describes the ozone hole progress distinguishing the monthly seasonality from the long-term evolution. Additionally, we present a timeline assessment of individual contribution of anthropogenic and natural chlorine and bromine species to Antarctic ozone loss during the 21$^{st}$ century, recognizing the independent impact arising from $LL^{Br}$ and $VSL^{Br}$ sources to the overall halogen-catalysed $O_3$ destruction.

## 2 Methods

The 3-D chemistry climate model CAM-Chem (Community Atmospheric Model with Chemistry, version 4.0)(Lamarque et al., 2012), included into the CESM framework (Community Earth System Model, version 1.1.1) has been used for this study. The model setup is identical to the CCMI-REFC2 experiment described in detail by (Tilmes et al., 2016), with the exception that the current setup includes a full halogen chemistry mechanism from the earth surface to the lower stratosphere

(Fernandez et al., 2014): i.e., instead of considering a constant lower boundary condition of 1.2 pptv for bromoform ($CHBr_3$) and dibromomethane ($CH_2Br_2$) or increasing $CH_3Br$ by 5 pptv, our model setup includes geographically-distributed and time-dependent oceanic emissions of six bromocarbons ($VSL^{Br}$ = $CHBr_3$, $CH_2Br_2$, $CH_2BrCl$, $CHBrCl_2$, $CHBr_2Cl$ and $CH_2IBr$) (Ordóñez et al., 2012). At the model surface boundary, zonally averaged distributions of long-lived halocarbons ($LL^{Cl}$ = $CH_3Cl$, $CH_3CCl_3$, $CCl_4$, CFC-11, CFC-12, CFC-113, HCFC-22, CFC-114, CFC-115, HCFC-141b, HCFC-142b and

$LL^{Br}$ = $CH_3Br$, H-1301, H-1211, H-1202 and H-2402), as well as surface concentrations of $CO_2$, $CH_4$, $H_2$, $N_2O$ are specified (Meinshausen et al., 2011). CAM-Chem was configured with a horizontal resolution of 1.9° latitude by 2.5° longitude and 26 vertical levels, from the surface up to ~40 km. To have a reasonable representation of the overall stratospheric circulation, the integrated momentum that would have been deposited above the model top is specified by an upper boundary condition (Lamarque et al., 2012). The model includes heterogeneous processes for active halogen species in polar stratospheric clouds

from MOZART-3 (Kinnison et al., 2007; Wegner et al., 2013). A full description of the CAM-Chem VSL configuration, detailing both natural and anthropogenic sources, heterogeneous recycling reactions, dry and wet deposition, convective uplift and large-scale transport has been given elsewhere (Ordóñez et al., 2012; Fernandez et al., 2014). This model configuration uses a fully-coupled Earth System Model approach, i.e. the ocean and sea-ice are explicitly computed.

Two ensembles of independent experiments (each of them with 3 individual ensemble members only differing in their 1950

initial atmospheric conditions) were performed from 1960 to 2100 considering a 10 years spin-up to allow for stratospheric circulation stabilization (i.e., each simulation started on 1950). The baseline setup ($run^{LL}$) considered only the halogen $LL^{Cl}$ and $LL^{Br}$ contribution from anthropogenic CFCs, halons and methyl chloride/bromide; while the second set of simulations





included, in addition to the $run^{LL}$ sources, the background biogenic contribution from $VSL^{Br}$ oceanic sources ($run^{LL+VSL}$). Differences between these two types of experiments allow quantifying the overall impact of natural $VSL^{Br}$ sources on stratospheric ozone. Please note that whenever we refer to "natural" contribution, we are pointing out to the contribution of biogenic $VSL^{Br}$ under a background stratospheric environment due to the dominant anthropogenic $LL^{Cl}$ and $LL^{Br}$ sources

(i.e., the natural fraction of long-lived chlorine and bromine are minor).

Unless stated otherwise, all figures were generated considering the ensemble average ($sim^{ens}$) of each independent experiment ($run^{LL}$ and $run^{LL+VSL}$), which in turn were computed considering the mean of the 3 independent simulations ($sim^{004}$, $sim^{005}$ and $sim^{006}$). For the case of vertical profiles and latitudinal variations, the zonal mean of each ensemble was computed to the monthly output before processing the data, while a Hamming filter with an 11 years window was applied to

10 all long-term time-series to smooth the data. Most of the figures and values within the text include geographically averaged quantities within the Southern Polar Cap (SP), defined as the region poleward of 63º S. For the case of the ozone hole area, we use the definition from the NASA Goddard Space Flight Center (GSFC), defined as the region with ozone columns below 220 DU located south of 40º S. Model results have been compared to the National Institute for Water and Atmospheric research – Bodeker Scientific (NIWA-BS) total column ozone database, which combines measurements from a

15 number of different satellite-based instruments (Bodeker et al., 2005).

## 3 Results and Discussion

### 3.1 Contribution of $LL^{Br}$ and $VSL^{Br}$ to stratospheric bromine

The 1960-2100 evolution of the stratospheric chlorine and bromine loading is shown in Fig. 1. The dominant anthropogenic $LL^{Cl}$ and $LL^{Br}$ scenarios included in our REFC2 simulations (Tilmes et al., 2016) show a pronounced peak at the end of the

20 20th century and beginning of 21st century, respectively, after which both their abundances decline. In comparison, the evolution of $VSL^{Br}$ sources remains constant in time, with a present-day fixed contribution of ~5 pptv (Ordóñez et al., 2012). Note that stratospheric $LL^{Cl}$ returns to its past 1980 levels before 2060, while the 1980 loading of $LL^{Br}$ is not recovered even by the end of the 21st century. Even when biogenic $VSL^{Br}$ sources remain constant, their relative contribution to the total bromine stratospheric loading changes with time: while for year 2000 $VSL^{Br}$ represents ~24% of total bromine, by the end of

25 the 21st century it reaches 40% of stratospheric bromine. These values are likely lower limits of the percentage contribution of biogenic sources to stratospheric bromine, as predicted increases on SST and oceanic nutrient supply are expected to enhance the biological activity and $VSL^{Br}$ production within the tropical oceans (Hossaini et al., 2012; Leedham et al., 2013). Furthermore, the increase in SST and atmospheric temperature projected for the 21st century, is expected to produce a strengthening of the convective transport within the tropics (Hossaini et al., 2012; Braesicke et al., 2013; Leedham et al.,

2013), which could additionally enhance the stratospheric injection of $VSL^{Br}$. The partitioning between carbon-bonded (source gas) and inorganic (product gas) bromine levels injected to the stratosphere are of great importance as they strongly





affect the ozone levels mostly in the lowermost stratosphere (Salawitch et al., 2005; Fernandez et al., 2014), which has implications at the altitudes where the strongest $O_3$-mediated radiative forcing changes due to greenhouse gases are expected to occur (Bekki et al., 2013).

## 3.2 Impact of $VSL^{Br}$ on the ozone hole evolution and its return date

The 1960-2100 evolution of the total ozone column within the southern polar cap ($TOZ^{SP}$, between 63ºS−90ºS) during October is illustrated in Fig. 2. Biogenic $VSL^{Br}$ introduce a continuous reduction in $TOZ^{SP}$ that exceeds the model ensemble variability between $run^{LL}$ and $run^{LL+VSL}$ experiments, and improves the overall model-satellite agreement (Fig. 2a). The temporal location of the minimum $TOZ^{SP}$ occurs simultaneously at the beginning of the 21$^{st}$ century in both experiments, with a minimum October mean $TOZ^{SP}$ column of 205 DU and 235 DU for $run^{LL+VSL}$ and $run^{LL}$, respectively. This leads to a

maximum October $TOZ^{SP}$ difference of −30 DU or ~14% of the overall $TOZ^{SP}$ during year 2003, while before 1970 the ozone differences remain practically constant and smaller than −14 DU, which represents only ~3.5% of the $TOZ^{SP}$. Analysis of the global annual column ($TOZ^{GB}$) between model experiments during the 1960-2100 interval shows approximately −3.6 DU difference, with maximum changes reaching −5.2 DU by year 1995. This represents < 2% of the annual $TOZ^{GB}$ observed for present time conditions and lies within the lower range of previous modelling studies for tropical and mid-latitudes over

the 1960-2005 period (Sinnhuber and Meul, 2015). These calculations reveal a much larger ozone loss efficiency of $VSL^{Br}$ on the Antarctic ozone layer than on global or tropical ozone stratospheric trends.

The different stratospheric bromine loading between $run^{LL+VSL}$ and $run^{LL}$ produces a different ozone column since the very beginning of the modelled period. The $\Delta TOZ^{SP}_{1980}$ (i.e. the difference with respect to 1980 baseline levels) during October shows a minimum for year 2003 of −92 DU and −77 DU for $run^{LL+VSL}$ and $run^{LL}$, respectively (Fig. 2b). Hence, from the 30

DU absolute difference shown in Fig. 2a, approximately half of the ozone offset is introduced by the background contribution of $VSL^{Br}$ on the global pre-ozone hole stratosphere. The additional ozone hole depletion (~15 DU by year 2000) induced by $VSL^{Br}$ is more noticeable between 1990 and 2010, i.e., when the stratospheric $LL^{Cl}$ loading also maximizes (see Fig. 1). This result is in agreement to Sinnhuber and Meul (2015), who reported a faster initial decrease and an overall better agreement between past mid-latitude $O_3$ trends and a model simulation forced with the additional contribution from $VSL^{Br}$

sources. Much smaller impacts are modelled on the 2$^{nd}$ quarter of the century when $LL^{Cl}$ constantly decreases and other ODS (such as $CH_4$ and $N_2O$) increase.

The vertical lines in Fig.2b indicate that the expected $TOZ^{SP}$ return date to October 1980 is approximately the same for both experiments: individual computations of the return date considering each of the independent ensemble members, show that the expected return date shift due to $VSL^{Br}$ lies within model uncertainties (Table 1), with mean ensemble values of ~(2052.7

± 0.7) for $run^{LL}$ and ~(2053.9 ± 4.8) for $run^{LL+VSL}$. In contrast, the maximum $TOZ^{SP}$ depletion observed for year 2000 increases by (−15.4 ± 12.4) DU when ~5 pptv of natural bromine are included, which exceeds the model internal variability. Thus, the Antarctic ozone hole return date, determined following the standard computation relative to the ozone column





existent in October 1980 (Eyring et al., 2010a, 2010b), is not significantly affected by the inclusion of natural $VSL^{Br}$ sources. This result contradicts the recent findings from Yang et al. (2014) and Oman et al. (2016), who estimated an increase between 7 to 10 years on the ozone hole return date. Note, however, that the former study performed non-coupled (without an interactive ocean) timeslice simulations including a speculative doubling of $VSL^{Br}$ sources on top of background $LL^{Cl}$ and

$LL^{Br}$ levels representative of years 2000 and 2050, while Oman et al. (2016) considered a single member climatic simulation for each type of experiment and thus lacks an assessment of the internal model variability. Our CAM-Chem results indicate that the inclusion of ~5 pptv of biogenic bromine does not only affect the future evolution of the ozone layer, but it reduces the overall background stratospheric ozone column prevailing in 1980. Hence, the additional depletion of $VSL^{Br}$ on ozone hole columns at their maximum depth shown in Fig. 2b considers the background impact of $VSL^{Br}$ chemistry on polar

stratospheric ozone throughout the 20th century, before and after the Antarctic ozone hole formed.

Agreement between model and observations for $TOZ^{SP}$ and $\Delta TOZ^{SP}_{1980}$ improves for all seasons when $VSL^{Br}$ are considered (Fig. 3). The maximum ozone difference between $run^{LL}$ and $run^{LL+VSL}$ is smaller than 10 and 5 DU for summer and fall, respectively, highlighting the much larger ozone depleting efficiency of the additional bromine from $VSL^{Br}$ sources during spring, when halogen chemistry dominates Antarctic ozone depletion. In all cases, the ozone return dates to 1980 seasonal

$TOZ^{SP}$ columns lay within the model uncertainties, with shorter return dates observed for the summer (~2045) and fall (<2040). Note also that the predicted springtime $\Delta TOZ^{SP}_{1980}$ will not return to their 1960 values by the end of the 21st century for neither $run^{LL}$ nor $run^{LL+VSL}$ simulations (Fig. 2b and Fig. 3). However, during fall positive $\Delta TOZ^{SP}_{1980}$ values are reached already by 2060, highlighting the different future seasonal behaviour of the Antarctic stratosphere (see Sect. 3.3).

### 3.2.1 Influence on the ozone hole area

We now turn to the effect of biogenic bromine on the Antarctic ozone hole area (OHA). Figure 4 indicates that the inclusion of $VSL^{Br}$ produces total ozone reductions larger than 10 DU from 1970 to 2070. This enhanced depletion extends well outside the limits of the southern polar cap (63ºS) and into the mid-latitudes (see grey line on Fig. 4). Most notably, the maximum ozone depletion driven by biogenic bromine is not located at the centre of the ozone hole but on the ozone hole periphery, close to the outer limit of the polar vortex (see polar views on Fig. 4). This result has implications for assessments

of geographical regions exposed to UV-B radiation: natural $VSL^{Br}$ leads to a total column ozone reduction between 20 and 40 DU over wide regions of the Southern Ocean near the bottom corner of South America and New Zealand.

Figure 5 indicates that the inclusion of $VSL^{Br}$ produces an extension of the maximum OHA of ~40% by the time where the maximum ozone hole is formed (2000th decade, 1995-2005), and it almost doubles the ozone hole extension during the 2030th decade (2025-2035). However, the inclusion of $VSL^{Br}$ drives a significantly lower impact on OHA by the time when

the ozone return date to October 1980 is expected to occur (2050th decade: 2045-2055). The agreement to the monthly mean ozone mass deficit (OMD) and OHA values obtained from the NIWA-BS database (Bodeker et al., 2005) is largely improved when $VSL^{Br}$ are considered. Most notably, the inclusion of $VSL^{Br}$ produces a maximum enlargement of the daily



OHA larger than 5 Million km$^2$, with a consequent enhancement of ~8 Million Tons on the OMD. Thus, the biogenic bromine-driven OHA enlargement is of equivalent magnitude, but opposite sign, to the chemical healing shrinkage estimated due to the current phase out of LL$^{Cl}$ and LL$^{Br}$ emissions imposed by the Montreal Protocol (Solomon et al., 2016).

Unlike the 1980-baseline ozone return date definition (which is normalized to a preceding but independent ozone column for
each ensemble), the OHA and OMD definitions are computed relative to a fixed value of 220 DU. Consequently, the $run^{LL+VSL}$ experiment shows larger ozone hole areas (white line on Fig. 4) and ozone mass deficits, but does not represent any significant extension on the size of the ozone hole by the time when the 1980-return date occurs. This supports the fact that the 1980-return date is controlled by the future evolution of the dominant LL$^{Cl}$ and LL$^{Br}$ sources. Note, however, that significant ozone depletion as large as −20 DU, and for latitudes as low as 60ºS, is still observed during 2060, i.e., after the
standard 1980-return date has been reached. This indicates that the contribution from VSL$^{Br}$ has significant implications on the baseline polar stratospheric ozone chemistry besides the above-mentioned impacts on ozone hole size, depth and return date.

### 3.2.2 Vertical distribution of the ozone hole depth

Timeline analysis of the mean October ozone vertical profile within the southern polar cap [O$_3(z)^{SP}$] is presented in Fig. 6.
Typically, the deepest O$_3(z)^{SP}$ reduction occurs at the lowermost stratosphere, i.e., between 200 and 100 hPa (~12 and 16 km), while during the pre- and post-ozone hole era, O$_3(z)^{SP}$ number densities peak between 100 and 50 hPa (~16 and 20 km). The additional O$_3(z)^{SP}$ depletion due to VSL$^{Br}$ sources is maximized precisely at the same altitudes where the minimum O$_3$ number densities are found: during the 2000$^{th}$ decade O$_3(z)^{SP}$ densities at 100 hPa for $run^{LL+VSL}$ and $run^{LL}$ are, respectively, <1.5 and <2.5 × 10$^{12}$ molecule cm$^{-3}$, which represents ~40% enhancement on the local ozone loss. This is in agreement to
the recent findings reporting that near-zero ozone concentrations in the deep Antarctic lower stratospheric polar vortex are only simulated when the VSL bromine sources are included (Oman et al., 2016). Interestingly, greater ozone loss is found in the periphery of the polar vortex, and below 25 hPa, due to the larger ozone number densities prevailing at those locations (see zonal panel on Fig. 6c). Above 25 hPa, O$_3(z)^{SP}$ is not significantly modified, with an overall VSL$^{Br}$ impact on ozone abundances smaller than 5%. This can be explained by the varying importance of bromine and chlorine chemistry at
different altitudes (see Sect. 3.4). Further analysis of Fig. 6d reveals that differences larger than 25% at ~100 hPa are only found between 1990 and 2010, confirming that the strongest impact of VSL$^{Br}$ sources occurs coincidentally with maximum LL$^{Cl}$ loadings (Fig. 1).

During the simulation period (i.e., 1960-2100), O$_3(z)^{SP}$ densities below 100 hPa are at least 10% lower for $run^{LL+VSL}$ than for $run^{LL}$. By year 2050, when the 1980 October return date is approximately expected to occur, the uppermost portion of the O$_3$
layer (above 50 hPa) shows strong signals of recovery and drives the overall TOZ$^{SP}$ return date, but the O$_3$ abundance below 50 hPa is still depleted relative to their pre-ozone hole era, mostly at high latitudes (Fig. 6d). It is only after year 2080 that the O$_3(z)^{SP}$ vertical profile is consistent with the pre-ozone hole period, although O$_3$ densities above 6 × 10$^{12}$ molec. cm$^{-3}$ are



still not recovered even by the end of the century (Fig. 6a,b). Between 2080 and 2100, inclusion of $VSL^{Br}$ still represents a 10% additional $O_3$ reduction at 100 hPa, which can be explained considering a shift from the predominant ozone destruction from chlorine to a bromine-driven depletion (whose efficiency is increased by the additional $VSL^{Br}$).

**3.3 Seasonal evolution of stratospheric Antarctic ozone**

Figures 7 show how the seasonal cycle of $TOZ^{SP}$ has changed during the modelled period, expanding from the typical solar-driven natural annual cycle prevailing in 1960 (Fig. 7a) to the strongly perturbed anthropogenic-induced cycle consistent with the formation of the Antarctic ozone hole (Fig. 7c, year 2000). $TOZ^{SP}_{July}$ normalizations on Figs. 7 and 8 have been computed respect to the $TOZ^{SP}$ value on July of each year, so the aperture, closure and depth of the ozone hole at each time is computed relative to the total ozone column prevailing during the preceding winter. Figure 8 shows the evolution of the annual cycle of $TOZ^{SP}$ as a function of simulated year for $run^{LL+VSL}$ and $run^{LL}$. During the pre-ozone hole era, the typical southern hemisphere natural seasonality is observed, with maximum October ozone columns for $run^{LL}$ that exceeds the values from $run^{LL+VSL}$ by ~5 DU. Starting on the seventies, the natural seasonal cycle is disrupted and the natural springtime maximum is replaced by a deep ozone reduction due to the ozone hole formation (Fig. 7b). The maximum $TOZ^{SP}_{July}$ difference respect to the preceding winter reach −95 DU for $run^{LL+VSL}$ (−75 DU for $run^{LL}$) during October 2000 (1995-2005 average), showing springtime differences greater than −30 DU (−20 DU) between September and December all the way from 1980 to 2050. The solid lines on Fig. 8 represent the temporal location of the monthly $TOZ^{SP}_{July}$ minimum for each simulation (white for $run^{LL+VSL}$ and black for $run^{LL}$). Starting on ~1981 the position of the $TOZ^{SP}_{July}$ annual minimum shifts from April (the radiatively driven fall minimum) to October (the springtime ozone hole minimum) for $run^{LL+VSL}$ (~1984 for $run^{LL}$). Accordingly, the returning of the $TOZ^{SP}$ annual minimum from October to April is delayed by ~4 years when $VSL^{Br}$ are considered (from 2047 for $run^{LL}$ to 2054 for $run^{LL+VSL}$). Table 2 shows the independent values for each of the independent ensemble members. Only if the baseline seasonal cycle is superposed below the long-term evolution of the polar stratospheric ozone layer (instead of considering the fixed normalization to October 1980), the inclusion of biogenic $VSL^{Br}$ introduces an extension on the ozone return date of ~(6.3 ± 12.2) years. Even though this value agrees with the estimations from Yang et al. (2014), it most probably represents a mere coincidence, as their timeslice computations only considered the changes in the maximum ozone hole depletion under different $VSL^{Br}$ loadings, while our analysis highlights the seasonal $TOZ^{SP}$ changes within a fully coupled climatic-simulation. Note, however, that in agreement to Table 1, the modelled delay on the return date computed considering the changes in the ozone seasonal cycle also lies within the internal model variability.

The dotted lines on Fig. 8 indicates the location of the double local $TOZ^{SP}_{July}$ maximums observed in Fig. 7b,d-e and allows determining how the timespan between the ozone hole formation and breaking for each year changes due to $VSL^{Br}$ chemistry. Between mid-1970s and mid-1980s, the seasonal development of the ozone hole for each year rapidly expanded shifting from a starting point as early as July through a closing date during the summer (December and January). Most





notably, the seasonal ozone hole extension during the 1$^{st}$ half of the century is enlarged as much as 1 month (from January to February) for $run^{LL+VSL}$ between 2020 and 2040. This occurs because the additional source of VSL$^{Br}$ produces a deepest October ozone minimum on top of the annual seasonal cycle, displacing the 2$^{nd}$ local maximum in between the minima to later dates (see Fig. 7D). During the 2000$^{th}$ decade, the location of the 2$^{nd}$ maxima, representing the closing end of the ozone hole, expands all the way to June of the following year because the ozone hole depletion during October is so large that its impacts persist until the following winter is reached: the year-round depletion of $TOZ^{SP}_{July}$ expands from 1990 to 2010 for $run^{LL}$, persisting ~7 years longer, from 1990 to 2017 for the $run^{LL+VSL}$ case. It is worth noting that because the ozone hole seasonal extension is not tied to a fixed TOZ value (as for example 220 DU) the ozone hole seasonal duration can be computed all the way to year 2100, even after the 1980-October standard ozone return date has already been achieved. These results indicate that even when LL$^{Cl}$ and LL$^{Br}$ will control the return date of the deepest ozone levels to the 1980-baseline value, the future evolution of VSL$^{Br}$ sources are of major importance to determine the future influence of halogen chemistry on the stratospheric Antarctic ozone seasonal cycle.

### 3.4 The role of chlorine and bromine ozone loss cycles ($ClOx^{LL}$ vs. $BrOx^{LL+VSL}$)

Bromine chemical cycles play a well-known role in the halogen-mediated springtime ozone hole formation (McElroy et al., 1986; Lee and Jones, 2002; Salawitch et al., 2005). Here we have used the same definition of odd-oxygen depleting families as in Table 5 from (Saiz-Lopez et al., 2014), with the exception of the iodine family which is not considered in this work. Figure 9 shows the temporal evolution of the percentage loss due to each cycle respect to the total odd-oxygen loss rate as well as the partitioning between the chlorine and bromine components within the halogen family. In the following, note that crossed $ClOx$-$BrOx$ cycles have been included into $BrOx^{LL+VSL}$ losses because both simulations include identical stratospheric LL$^{Cl}$ loading but a ~5 pptv difference in total bromine (see Fig. 1).

Between approximately 1980 and 2060 the dominant ozone depleting family within the springtime Antarctic ozone hole is halogens: $ClOx^{LL} + BrOx^{LL+VSL}$ surpass the otherwise dominant contribution from $NOx$ and $HOx$ cycles (Fig. 9A): e.g., during the year of largest ozone depletion (i.e. October 2003), halogens represent more than 90% of the total odd oxygen loss at 100 hPa, while $NOx$ and $HOx$ cycles contribute ~5% and less than 2%, respectively. By year 2050, when the 1980-October baseline ozone return date is expected to occur, the overall $BrOx^{LL+VSL}$ cycles represent ~45% of the total ozone loss by halogens occurring at 100 hPa (Fig. 9B) and ~35% when integrated in the stratosphere (Fig. 9C). Even though $ClOx^{LL}$ losses represent as much as 80 % of the halogen-mediated ozone loss during the 2000$^{th}$ decade, the additional contribution from VSL$^{Br}$ drives bromine chemistry ($BrOx^{LL+VSL}$) to dominate ozone loss by halogens approximately by year 2070. The contribution of $BrOx^{LL+VSL}$ cycles to ozone loss was higher than $ClOx^{LL}$ also before 1975, i.e. before and during the fast increase in anthropogenic CFCs occurred (Fig. 9B). This implies that, although anthropogenic chlorine has controlled and will control the long-term evolution of springtime stratospheric ozone hole, its overall depleting potential in the lowermost stratosphere is strongly influenced by the total (natural + anthropogenic) stratospheric inorganic bromine, with a non-





negligible contribution (up to ~30%) from the biogenic $VSL^{Br}$ oceanic sources. Within the $run^{LL}$ experiment, $BrOx^{LL}$ cycles never surpass the contribution of $ClOx^{LL}$ losses, revealing the significant enhancement of inter-halogen $ClOx^{LL}$-$BrOx^{LL+VSL}$ depletion due to the additional source of natural $VSL^{Br}$.

There is a clear variation on the height at which $ClOx$ and $BrOx^{LL+VSL}$ cycles produces its maximum destruction, as well as the period of time when the losses by each family dominate respect to the others. For example, pure $ClOx^{LL}$ cycles account for more than 80% of the total halogen losses above 10 hPa during the whole 21$^{st}$ century, while $BrOx^{LL+VSL}$ cycles maximize close to the tropopause. Figure 10 shows that during the Antarctic spring, stratospheric bromine chemistry below 50 hPa has been at least as important as chlorine before and after the ozone hole era. Thus, the future evolution of stratospheric $LL^{Cl}$ levels will control the ozone hole return date, but the role played by $VSL^{Br}$ by that time will be as large as the one arising from $LL^{Br}$. This effect will be most evident within the lower stratospheric levels: bromine is globally ~60 times more efficient than chlorine in depleting ozone (Daniel et al., 1999; Sinnhuber et al., 2009), but its efficacy relies mostly on the background levels of stratospheric chlorine and the prevailing temperature affecting the rate of the inter-halogen crossed reactions (Saiz-lopez and Fernandez, 2016). Additionally, the extent of $ClOx^{LL}$ depletion within the Antarctic vortex relies on the occurrence of heterogeneous activation of chlorine reservoir species on polar stratospheric clouds, which in turn depend on ambient temperature. Then, the efficiency of $BrOx^{LL+VSL}$ depleting cycles relative to chlorine is reduced in the colder lower stratosphere at high latitudes during the 2000$^{th}$ decade (see lower panels on Fig. 10), while the $BrOx^{LL+VSL}$ contribution is larger at mid latitudes and increase in importance as we move forward into the future.

The representation of the $ClOx^{LL}$ and $BrOx^{LL+VSL}$ contributions shown in Fig. 11 allows addressing two interesting features related to the seasonal and long-term evolution of lower stratospheric Antarctic ozone. For any fixed year during the ozone hole era, bromine chemistry reaches a minimum during austral spring, while it increases during the summer and fall months. For example, the $BrOx^{LL}$ contribution to total halogen loss at 100 hPa for year 2000 is 25% during October, 65% in December and greater than 80% by March. Thus, if the Antarctic return date delay is computed considering the baseline 1980 value for the fall months, a greater impact from $VSL^{Br}$ is observed (see Fig. 3c). Accordingly, the evaluation of the long-term impact of $ClOx^{LL}$ and $BrOx^{LL+VSL}$ cycles on the evolution of Antarctic ozone changes abruptly if we focused on the fall months instead of considering the October mean. In the lower stratosphere, chlorine chemistry is dramatically enhanced during October due to the formation of the Antarctic ozone hole, but during summer and fall $ClOx^{LL}$ losses decrease, representing less than 20% of the total halogen loss (March mean) during the 21$^{st}$ century.

## 4 Discussion and Concluding Remarks

We have shown that biogenic $VSL^{Br}$ have a profound impact on the depth, size and vertical distribution of the springtime Antarctic ozone hole. The inclusion of $VSL^{Br}$ improves the quantitative 1980-2010 model/satellite agreement of $TOZ^{SP}$, and it is necessary to capture the lowest October mean ozone hole values. Our model results also show that, even when the





maximum springtime depletion is increased by the inclusion of $VSL^{Br}$, the future recovery of Antarctic ozone to the prevailing levels before 1980 is primarily driven by the evolution of the dominant $LL^{Cl}$ and $LL^{Br}$ sources: i.e. $VSL^{Br}$ sources does not change significantly the estimated return date. This can be explained considering the larger impact of bromine chemistry during periods of high inorganic chlorine loading, as well as due to the background impact of the additional

bromine on the past global stratosphere. Other chemistry climate modelling studies estimated a decade enlargement of the expected return date based on a single member simulation (Oman et al., 2016), but those studies considered an approximate $VSL^{Br}$ approach increasing the $CH_3Br$ lower boundary condition by ~5 pptv, while here we performed 6 independent simulations including geographically-distributed time-dependent $VSL^{Br}$ oceanic sources. Note, however, that free-running ocean interactive simulations as the ones performed in this work possess a very large model internal variability, so more

ensemble members might be required to better address the important issue of the return date. The $TOZ^{SP}$ minimum and the ozone hole depth in the lower stratosphere are both increased by 14% and 40%, respectively, when $VSL^{Br}$ is considered. This effect is more pronounced in the periphery of the ozone hole and within the heights of smaller ozone densities. Interestingly, biogenic bromine produces an enlargement of the OHA of 5 million $km^2$, equivalent to that of the recently estimated Antarctic ozone healing due to the implementation of the Montreal Protocol. This large effect of oceanic $VSL^{Br}$ on the OHA

highlights the importance of including biogenic bromine in climate assessments of the future Antarctic ozone layer. As the anthropogenic emissions of $LL^{Cl}$ and $LL^{Br}$ are projected to decrease in the future following the Montreal protocol, the natural $VSL^{Br}$ relative contribution will represent as much as 40% of stratospheric bromine throughout the 21st century, or even more if the oceanic $VSL^{Br}$ source strength and deep convection tropical injection increase in the near future (Hossaini et al., 2012; Leedham et al., 2013). Indeed, enhanced bromine $BrOx^{LL+VSL}$ cycles will dominate the chemistry of the lowermost

stratosphere over Antarctica before a complete recovery of the global ozone layer from $LL^{Br}$ and $LL^{Cl}$ has occurred. Hence, oceanic $VSL^{Br}$ possess leverage to significantly influence the future evolution of the Antarctic ozone layer.

**Acknowledgments**

We would like to thank Greg Bodeker of Bodeker Scientific, funded by the New Zealand Deep South National Science

Challenge, for providing the combined total column ozone database. This work was supported by the Consejo Superior de Investigaciones Científicas (CSIC), Spain. The National Center for Atmospheric Research (NCAR) is funded by the National Science Foundation NSF. Computing resources (ark:/85065/d7wd3xhc) were provided by the Climate Simulation Laboratory at NCAR's Computational and Information Systems Laboratory (CISL), sponsored by the NSF. The CESM project (which includes CAM-Chem) is supported by the NSF and the Office of Science (BER) of the U. S. Department of

Energy. R.P.F. would like to thanks Pablo Cremades for his technical help on post-processing the output data, and to CONICET, FCEN-UNCuyo and UTN-FRMendoza for financial support.





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





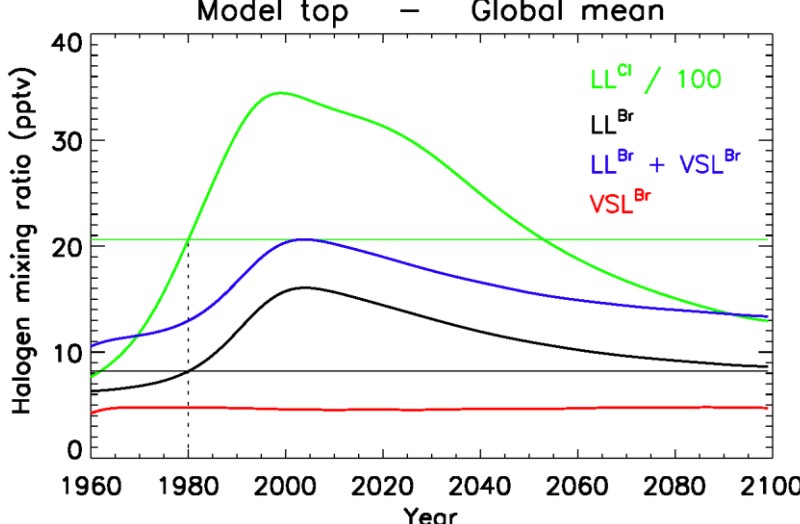

**Figure 1: Temporal evolution of the annual mean global stratospheric halogen loading at the top of the model (i.e., 5 hPa) for long-lived chlorine (LL$^{Cl}$) and bromine (LL$^{Br}$), as well as very short-lived bromine (VSL$^{Br}$). The horizontal lines indicate the LL$^{Cl}$ and LL$^{Br}$ mixing ratio for year 1980. LL$^{Cl}$ mixing ratios have been divided by 100.**




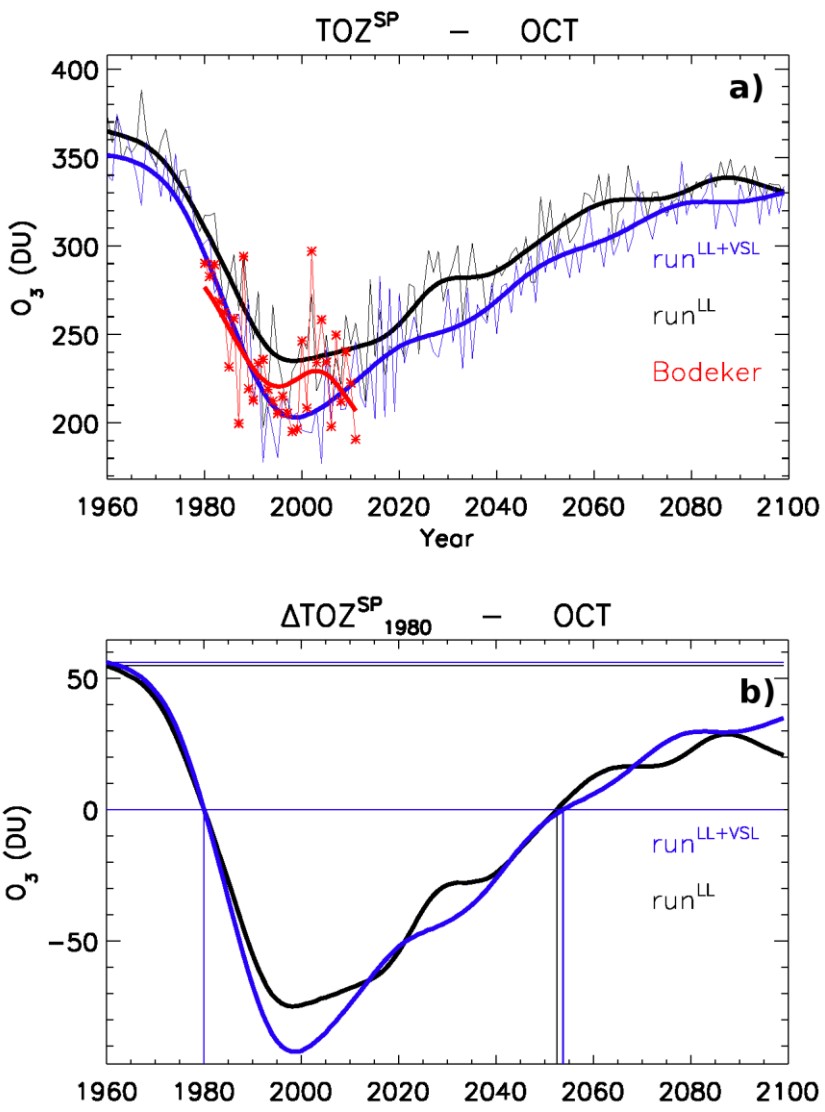

**Figure 2: Temporal evolution of the total ozone column averaged within the southern polar cap (TOZ$^{SP}$) during October. CAM-Chem results are shown in blue for *run*$^{LL+VSL}$ and black for *run*$^{LL}$. A) Absolute TOZ$^{SP}$ values for the ensemble mean (thin lines) and the 11-years smooth timeseries (thick lines). Red lines and symbols show merged satellite and ground base measurements from the Boedeker database averaged within the same spatial and temporal mask as the model output. B) Total ozone column adjusted respect to October 1980 ($\Delta$TOZ$^{SP}_{1980}$ = TOZ$^{SP}_{year}$ − TOZ$^{SP}_{1980}$). The zero horizontal line indicates the October $\Delta$TOZ$^{SP}_{1980}$ column for each experiment, while their respective return dates to 1980 are shown by the vertical lines. The upper horizontal lines represent the TOZ$^{SP}$ column during October 1960 for *run*$^{LL+VSL}$ and *run*$^{LL}$.**




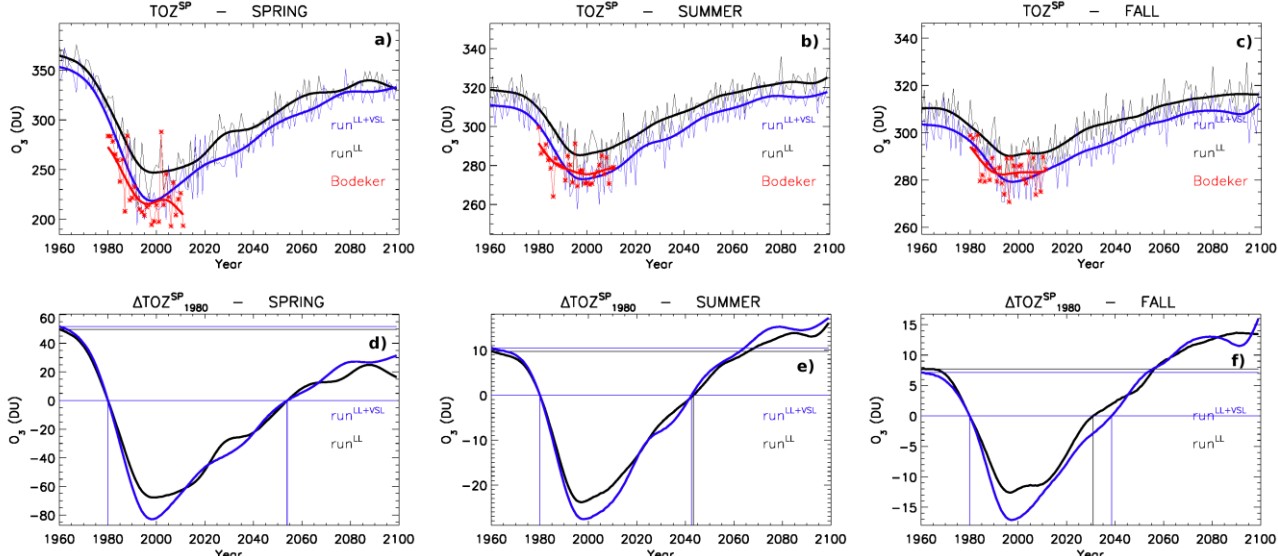

**Figure 3: Idem to Fig. 2, but computing the 3-month average for Spring (AUG-SEP-OCT, left), Summer (DEC-JAN-FEB, middle) and Fall (MAR-APR-MAY, right).**





**Figure 4: Ensemble mean of temporal evolution of Southern Hemisphere October TOZ as a function of latitude for A)** $run^{LL+VSL}$**; B)** $run^{LL}$**; C) absolute difference between** $run^{LL+VSL}$ **and** $run^{LL}$**; and D) percentage difference between experiments. The double inset on the bottom of each panel shows the October TOZ mean polar view during the 2000 (1995-2005 mean, left) and 2030 (2025-2035 mean, right) decade. The solid lines on each panel show the** $O_3 = 220$ **DU limit defining the ozone hole area (GSFC, NASA) for each simulation (white for** $run^{LL+SL}$ **and black for** $run^{LL}$**), while the solid grey line show the 63ºS parallel defining the Southern Polar cap (SP) over which TOZ$^{SP}$ is computed.**





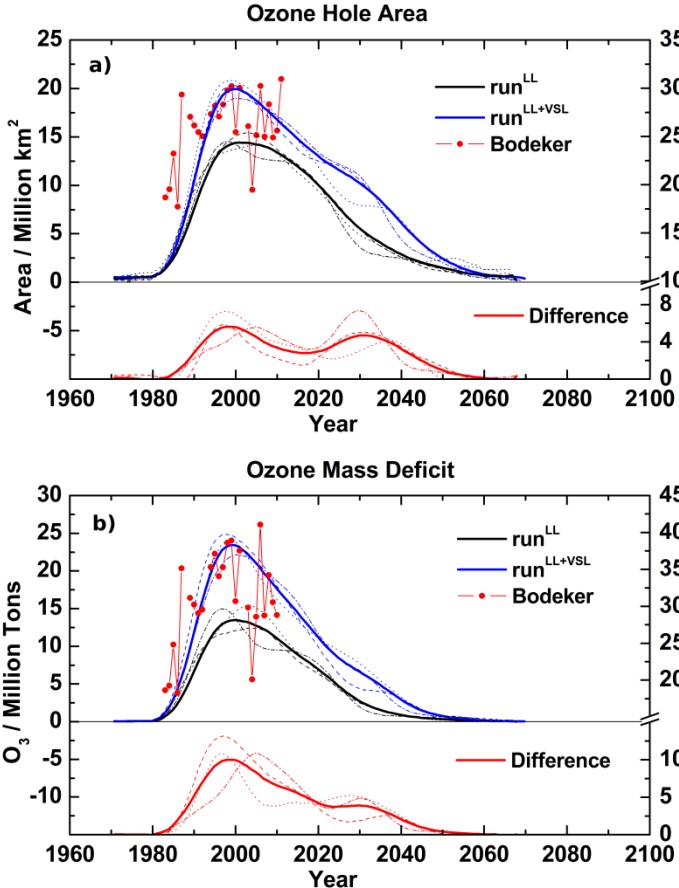

**Figure 5: Temporal evolution of the ozone hole area (A) and ozone mass deficit (B) for both experiments (black for $run^{LL}$ and blue for $run^{LL+VSL}$) on the left axis, as well as the difference between runs (red) on the right axis. Solid thick lines show the ensemble mean for each experiment; while the dashed, dotted and dashed-dotted thin lines correspond to each of the 3 independent simulations ($sim^{004}$, $sim^{005}$ and $sim^{006}$) for each run.**





**Figure 6: Temporal evolution of the ozone vertical profile averaged within the South Polar Cap ($O_3(z)^{SP}$) for the month of October for $run^{LL+VSL}$ (panel A); $run^{IL}$ (panel B); the absolute difference between experiments (panel C); and the percentage difference (panel D). The double inset on the bottom of each panel shows the October zonal mean vertical distributions during the 2000 (1995-2005 mean, left) and 2030 (2025-2035 mean, right) decades. All panels show ozone number densities (i.e., molec cm$^{-3}$) to highlight its contribution to the overall TOZ column. The lower solid line (white for $run^{LL+VSL}$ and black for $run^{LL}$) indicates the location of the tropopause, while the higher solid line indicates the height where $O_3$ number density equals its value at the tropopause.**



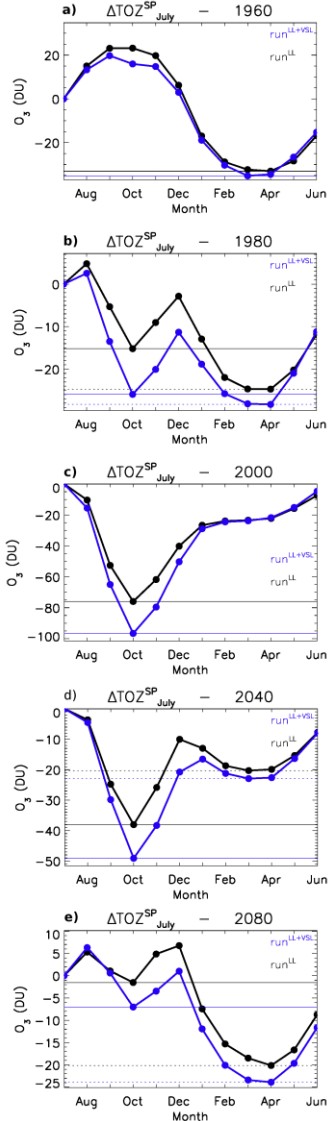

**Figure 7: Seasonal variation of $\Delta TOZ^{SP}_{July}$ for $run^{LL+VSL}$ (blue) and $run^{LL}$ (black) ensemble means at different years: A) 1960, before the ozone first appeared; B) 1980, where the appearance of the ozone hole produces a small $TOZ^{SP}$ local minimum during spring; C) 2000, when the ozone hole depth in October maximize; D) 2040, when $TOZ^{SP}$ minimum still appears in spring during the ozone hole recovery timeline; E) 2080, after the $TOZ^{SP}$ global minimum has already returned to fall into its natural seasonal cycle. The solid and dashed horizontal lines highlight the local and global $TOZ^{SP}$ minimum for each experiment. $\Delta TOZ^{SP}_{July}$ baseline adjustment have been computed relative to the modelled $TOZ^{SP}$ in July of the preceding winter for each year ($\Delta TOZ^{SP}_{July}$ = $TOZ^{SP}_{Time}$ − $TOZ^{SP}_{July}$).**




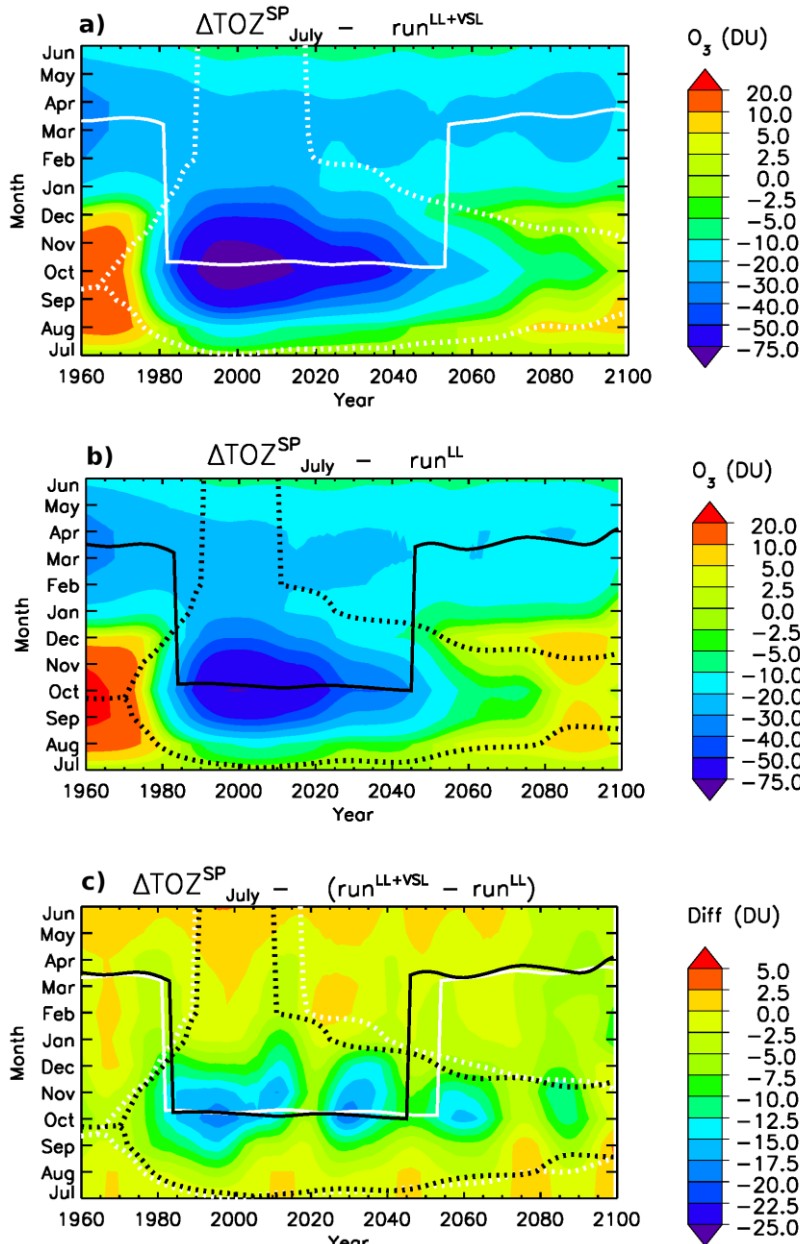

**Figure 8: Evolution of $\Delta TOZ^{SP}_{July}$ as a function of the year and month. A)** $run^{LL+VSL}$ **ensemble mean; B)** $run^{LL}$ **ensemble mean; and C) Absolute difference between the simulations.** $\Delta TOZ^{SP}_{July}$ **baseline adjustment have been computed relative to the modelled** $TOZ^{SP}$ **in July of the preceding winter for each year (**$\Delta TOZ^{SP}_{July} = TOZ^{SP}_{Time} - TOZ^{SP}_{July}$**). The solid line indicates the location of the** $TOZ^{SP}$ **annual minimum for each ensemble (white for** $run^{VSL}$ **and black for** $run^{noVSL}$**), while the dashed lines indicate the shifts on the** $TOZ^{SP}$ **local maximums arising on each side of the springtime minimum (see Fig. 7).**



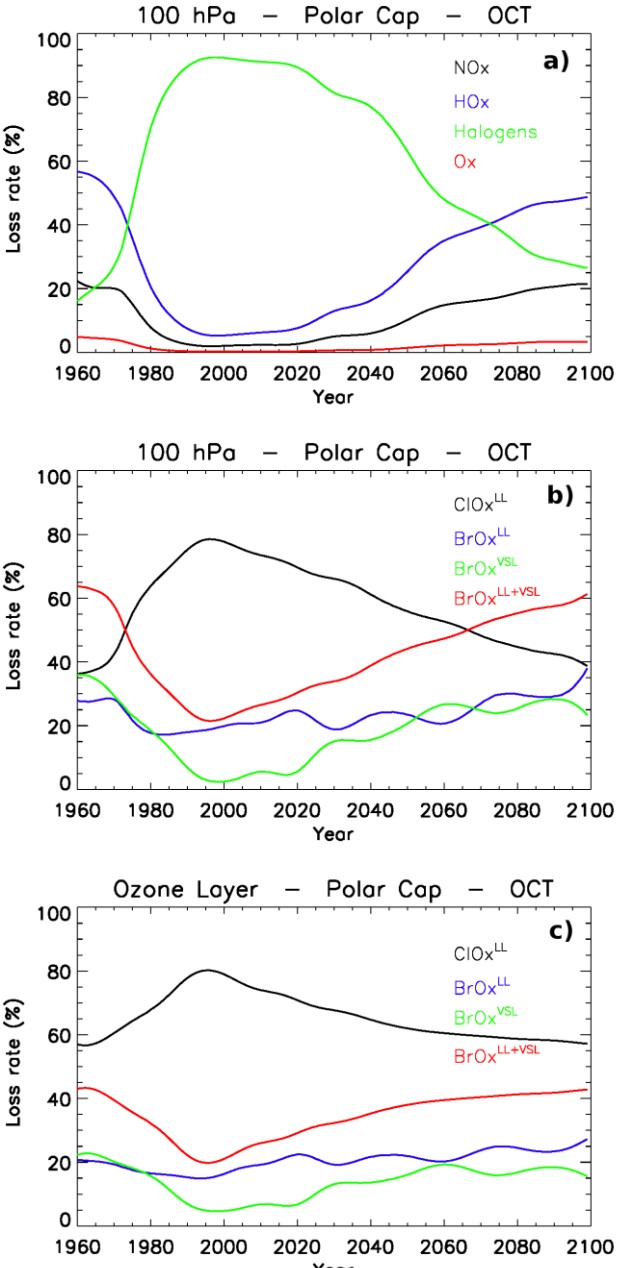

**Figure 9: Temporal evolution of the October mean odd-oxygen loss rates within the Southern Polar cap. A) Percentage contribution of each ozone depleting family (HOx, NOx, Ox and Halogens) respect to the total loss rate at 100 hPa (~15 km); B) percentage contribution of each halogen family (ClOx, BrOx$^{LL}$, BrOx$^{VSL}$, and BrOx$^{LL+VSL}$) respect to the whole halogen loss rate at 100 hPa; and C) Idem to panel B) but vertically integrated within the lower stratosphere (i.e., in-between the white lines shown in Fig. 6). Ensemble mean values are shown.**

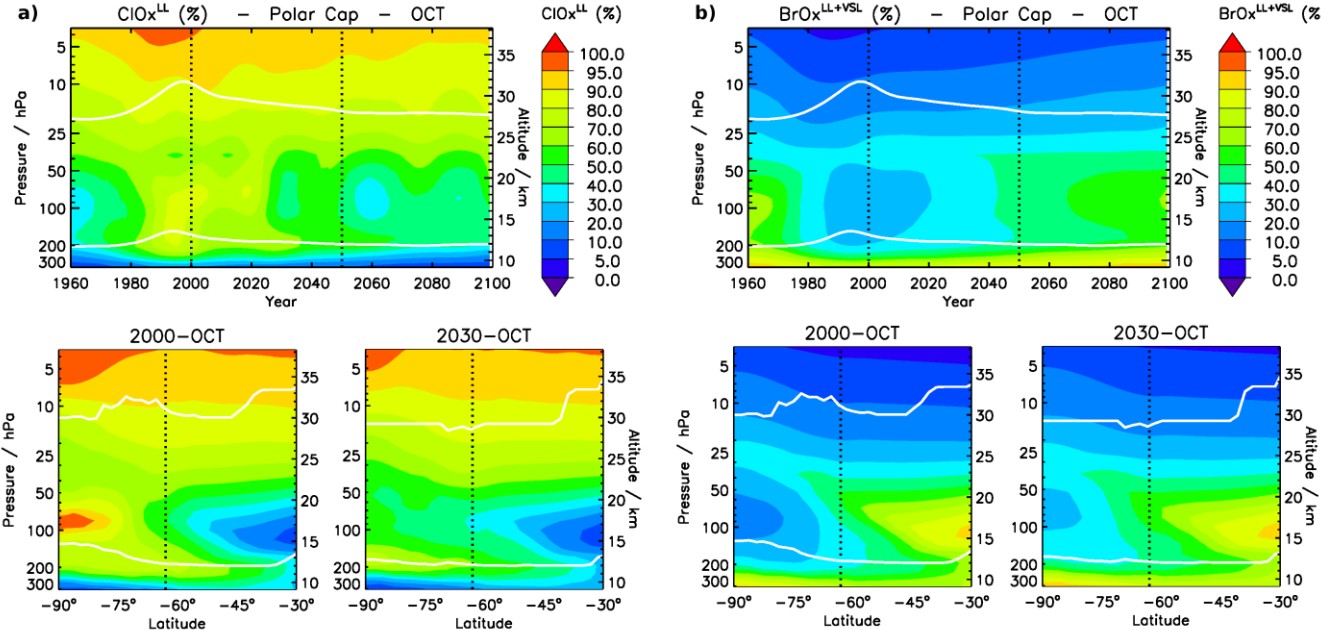

**Figure 10:** Evolution of the odd-oxygen loss rate vertical profiles (VP) within the South Polar Cap. The percentage contribution of each family respect to the whole halogen loss during October is shown for A) the ClOx$^{LL}$ family; and B) the BrOx$^{LL+VSL}$ family. The inset below each VP shows the October zonal mean vertical distributions of odd-oxygen losses during the 2000 (1995-2005 mean, left) and 2030 (2025-2035 mean, right) decades. All results are for the $run^{LL+VSL}$ ensemble. The lower solid white line indicates the location of the tropopause, while the higher solid line indicates the height where O$_3$ number density equals its value at the tropopause.





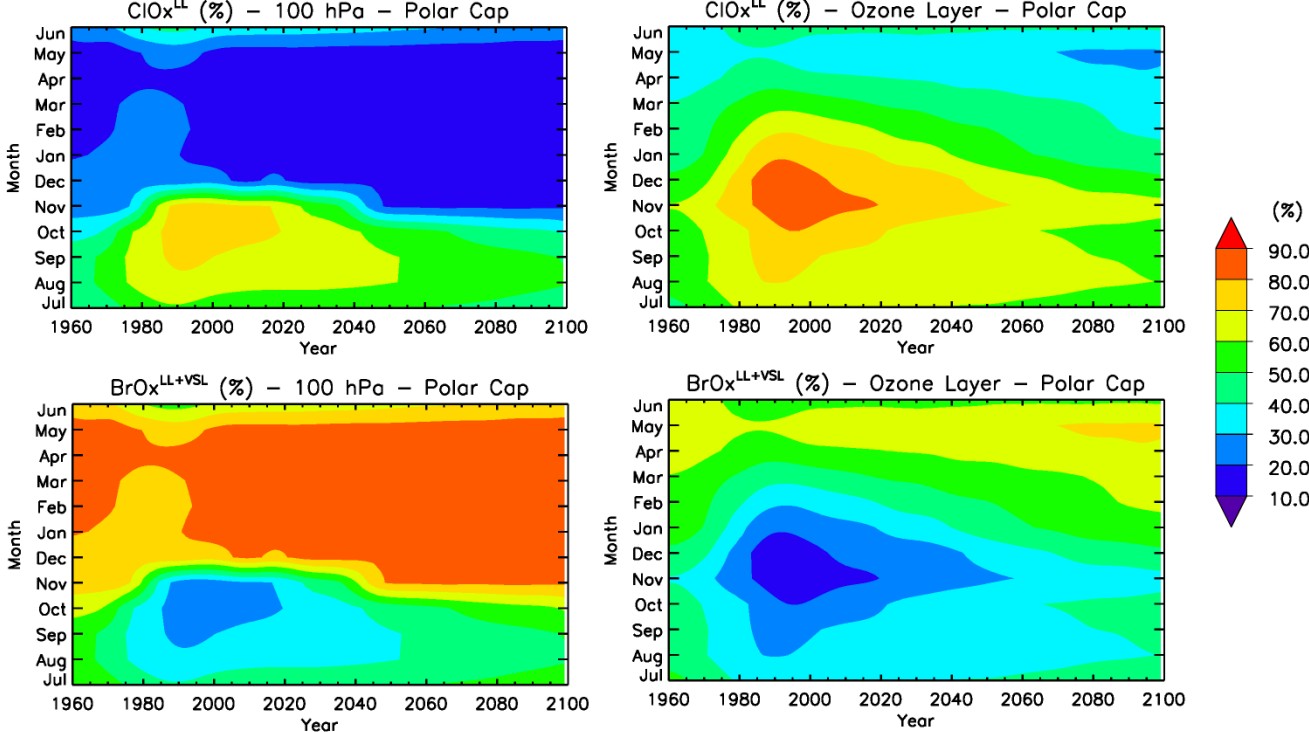

**Figure 11:** Evolution of the halogen-catalysed Odd-Oxygen loss rates as a function of the year and month for the $ClO_x^{LL}$ family (top row) and he $BrOx^{LL+VSL}$ family (bottom row). The left column show loss rate values at 100 hPa (~15 km), while in the right column the loss rates have been vertically integrated within the lower stratosphere (i.e., in-between the white lines shown in Fig. 10). Results are for the $run^{LL+VSL}$ ensemble.





**Table 1: Estimation of the ozone return date, minimum ozone column within the Southern Polar Cap ($TOZ^{SP}_{min}$) and the maximum ozone hole area ($OHA_{max}$) modelled with CAM-Chem for different simulations and ensemble members.**

| | Return date $^{1980}$ (years) | | $TOZ^{SP}_{min}$ (DU) | | $OHA_{max}$ (Million km$^2$) | |
|---|---|---|---|---|---|---|
| | $run^{LL+VSL}$ | $run^{LL}$ | $run^{LL+VSL}$ | $run^{LL}$ | $run^{LL+VSL}$ | $run^{LL}$ |
| $sim^{004}$ | 2058.9 | 2053.4 | −88.9 | −72.8 | 19 | 14.2 |
| $sim^{005}$ | 2053.4 | 2052.2 | −98.1 | −72.8 | 20.8 | 13.8 |
| $sim^{006}$ | 2049.3 | 2052.3 | −90.7 | −85.8 | 20.3 | 15 |
| ensemble | 2053.9 ± 4.8 | 2052.7 ± 0.7 | −92.6 ± 4.9 | −77.2 ± 7.5 | 20.0 ± 0.9 | 14.3 ± 0.6 |
| Shift | (1.2 ± 5.5) | | (−15.4 ± 12.4) | | (5.7 ± 1.5) | |

**Table 2: Estimation of the period of time where the annual minimum $\Delta TOZ^{SP}_{July}$ is observed during Spring for different simulations and ensemble members.**

| | $run^{LL+VSL}$ | | $run^{LL}$ | |
|---|---|---|---|---|
| | Start date (year) | Return date (year) | Start date (year) | Return date (year) |
| $sim^{004}$ | 1982 | 2057 | 1983 | 2053 |
| $sim^{005}$ | 1982 | 2049 | 1984 | 2049 |
| $sim^{006}$ | 1981 | 2048 | 1985 | 2040 |
| ensemble | 1981.6 ± 0.6 | 2051.3 ± 4.8 | 1984.0 ± 1.0 | 2047.3 ± 6.6 |
| Period (years) | (69.6 ± 4.6) | | (63.3 ± 7.6) | |
| Shift | (6.3 ± 12.2) | | | |

