# Peer review of "Impact of biogenic very short-lived bromine on the Antarctic ozone hole during the 21st century"

_Atmospheric Chemistry and Physics, 2016_

## Referee Comment (RC1) · Anonymous Referee #1 · 4 Nov 2016

This study examines the impact of VSL Br on stratospheric ozone depletion in the CAM-Chem model using multiple ensemble members including a coupled ocean. Finding better agreement with observations when the impact is included in the model but not finding any significant delay in the Antarctic ozone return date. Also, this work finds an increasingly important effect of biogenic bromine on the future Antarctic ozone layer. Overall I find the paper clear and well written and of interest to the ACP community, however, I do have strong concerns about the coarseness of the representation of the stratosphere in the model used and would appreciate the authors addressing these concerns or clearly stating the uncertainties that this may cause in their conclusions. I do appreciate the explicit representation of the bromocarbons, interactive ocean, and

multiple ensembles used in this study but they still all rely on confidence in the representation of the stratosphere and its response to the forcing applied.

The CAM-Chem model used in this study has 26 vertical levels and a model top around ~40km and in fig 1 state the top model level is around 5 hPa. Please add to the model description how many levels are above the tropopause. Typically models of this coarse vertical resolution have less than a dozen or so levels above the tropopause.

Have you done any comparisons to a model with a well resolved stratosphere like WACCM with respect to circulation, mean age, PSC area, or ClOx, BrOx, NOx, HOx concentrations? That might help to quantify uncertainties or to understand the extent that a model with so few stratospheric levels can simulate or properly represent these important quantities.

Recovery of Antarctic October ozone to 1980 levels occurs in the mid 2050s in the CAM-Chem simulations this is significantly earlier than the 4 models used in the WMO 2014 assessment which returned in the 2070s - 2080s (fig 3-15). These models had well resolved stratospheres and were evaluated in CCMVal-2 to have the best representation of stratospheric transport and chemistry. Why should we have confidence in the earlier recovery estimate from CAM-Chem or can you appropriately caveat the conclusions made with this uncertainty?

Page 3 lines 22-24 I couldn't find a figure in Lamarque et al. 2012 that shows reasonable overall stratospheric circulation from including the integrated momentum flux that needs to be in a model with such a low upper boundary. Can you cite or include figures that compare these simulation to observations of mean age or other measures of stratospheric circulation or transport? How is this handled in the future is it interactive or fixed. Is the circulation change over time comparable to models with a well resolved stratosphere.

Could you explain in the paper with a model top at around 5 hPa (from figure 1) how do you represent the 5-8% of total column ozone above the model top?

Can you show or discuss how much Br goes through the tropical tropopause in these two sets of simulations are they consistent with published aircraft and satellite estimates when VSL Br is accounted for. How well is polar BrO columns modeled compared to observations in CAM-Chem.

Figure 3 Why is Spring Aug.-Oct. rather than SON

On Figure 5 there appears to be a significant difference in the early 1980s in ozone hole area between the observations and CAM-Chem simulations but I didn't see this mentioned in the text. Would you expect an underestimation of ozone hole area to be significant to the earlier return date found in CAM-Chem. How does this impact your conclusions?

Figure 6 There appears to be large 30-year time scale variability in the polar cap ozone in the ensemble average is this coming from the ocean, can you explain. The panels with the time axis show dotted lines at 2000 and 2050 but if the label is correct on the other panels you are meaning to highlight 2030 instead. Same on figure 4 and fig 10.

Page 2 lines 3-5 when discussing the Antarctic ozone return dates you reference the older CCMVal-2 and WMO 2010 results and not the more recent WMO 2014 which had a significantly later recovery estimate, please add mention of the WMO 2014 result here.

Page 3 lines 13-15 For readers unfamiliar with CCMI-REFC2 can you state the GHG and ODS scenario used in this study.

Page 4 lines 13-15 for the total column ozone database please state which version used (is it the latest) and what years it covers. If it continues through 2015 can figures 2 and 3 be extended to include more recent years.

Page 8 line 19-20 4 years doesn't agree with difference 2047 and 2054 in the text. I think you meant to write 2051 instead of 2054.

Given that the largest differences were found in the periphery of the ozone hole does

the definition used 63-90 vs 60-90 make any difference in your dates. I have seen both regions used so either is fine, I would just suggest checking that it doesn't make a difference.

Page 9 line 2 change deepest to deep Page 9 line 17 change "respect to" to "with respect to" Page 10 line 5 same as above

---

## Referee Comment (RC2) · B.-M. Sinnhuber (Referee) · 28 Nov 2016

The study by Fernandez et al. applies the CAM-Chem chemistry climate model to investigate the impact of oceanic emissions of very short-lived brominated source gases (VSLS_Br) on the Antarctic ozone hole during the 21st century. This is a very thorough and well performed study and the paper is well written. Its analyses help to further understand results of previous studies and demonstrate the importance of oceanic VSL_Br emissions for stratospheric ozone. I suggest publication in Atmos. Chem. Phys. after consideration of the following comments.

Specific comments

[Figure]

For the comparison to the results of the recent study by Oman et al., Table 1 is revealing, showing a difference of almost 10 years in return dates for different ensemble members using the same boundary conditions. (Hope I understood this correctly.) I suggest to make this point even clearer when discussing the differences to Oman et al.

The effect of VSLS_Br maximizes in the late 1990s (e.g., Figs 4c and 6c), but there is a secondary maximum around 2030 (and following minor mixima around 2060 and 2090). Why is that? Is this an artifact from the 11-year smoothing?

p2,l20: Reference to Sinnhuber and Meul might be slightly misleading: They showed, that indeed the highest impact is during periods of high aerosol loading, but the strongest impact on ozone depletion is not at mid-latitudes, but at the Antarctic ozone hole.

p4,l28: "...the increase in SST and atmospheric temperature...is expected to ...additionally enhance the stratospheric injection of VSL_Br": This effect should already be included in the current simulations, so would not be additional, as I understand?

p5, ozone hole evolution: Do the model simulations include volcanic eruptions or not? Would be good to mention during the discussion of Fig.2, as Pinatubo may have played a role.

Fig. 3: The separation into different seasons is very helpful, but why is spring defined as AUG-SEP-OCT, instead of SEP-OCT-NOV, and why not include winter (JUN-JUL-AUG) for completeness?

p11,l18: "...or even more if the oceanic VSL_Br source strength and deep convection increases...": For the deep convection, I assume this is already considered here (see my comment above), while it should be acknowledged that the increase in oceanic source strength is largely speculative at this point.

Technical corrections:

Sometimes reference is to Carpenter et al., 2014, sometimes to WMO, 2014 (e.g., p2,l10) with no obvious reason for the distinction.

p2,l13: Saiz-lopez -> Saiz-Lopez

p3,l31: "on 1950" -> "in 1950"

p7,l15: "at the lowermost" -> "in the lowermost" (?)

---

## Author Comment (AC1) · 26 Dec 2016

**ANSWERS TO REVIEWERS**

**Impact of biogenic very short-lived bromine on the Antarctic ozone hole during the 21[st] century**

**R. P. Fernandez[1,2], D. E. Kinnison[3], J-F. Lamarque[3] , S. Tilmes[3] and A. Saiz-Lopez[1]**

**General Answer**

We are very grateful to anonymous Reviewer_ #1 and Bjoern-Martin Sinnhuber for their constructive comments and suggestions, which helped us to improve the manuscript. In the present revised version we have fully addressed all the reviewer's comments, including updates on references, clarifying descriptions of model configurations, validation of model performance, rephrasing of misleading implications and introducing corrections on tables and figures. We've also prepared a supporting document to be included as Supplementary Material, which summarizes the main responses given to the reviewers and complement the results presented in the main text.

To facilitate the reading, the original comments made by the reviewers have been copy-pasted here using **bold font**, while our answers are given in regular font. Additionally, we have copied into this response letter the current changes made to the original manuscript, using a *blue (corrected text)* and/or *italic (original text)* font type.

\*\*\*\*\*\*\*\*\*\*\*\*\*\*\*\*\*\*\*\*\*\*\*\*\*\*\*\*\*\*\*\*\*\*\*\*\*\*\*\*\*\*\*\*\*\*\*\*\*\*\*\*\*\*\*\*\*\*\*\*\*\*\*\*\*\*\*\*\*\*\*\*\*\*\*\*\*

**1   Anonymous Reviewer_#1**

\*\*\*\*\*\*\*\*\*\*\*\*\*\*\*\*\*\*\*\*\*\*\*\*\*\*\*\*\*\*\*\*\*\*\*\*\*\*\*\*\*\*\*\*\*\*\*\*\*\*\*\*\*\*\*\*\*\*\*\*\*\*\*\*\*\*\*\*\*\*\*\*\*\*\*\*\*

**1.1   General Remarks**

**This study examines the impact of VSL Br on stratospheric ozone depletion in the CAMChem model using multiple ensemble members including a coupled ocean. Finding better agreement with observations when the impact is included in the model but not finding any significant delay in the Antarctic ozone return date. Also, this work finds an increasingly important effect of biogenic bromine on the future Antarctic ozone layer. Overall I find the paper clear and well written and of interest to the ACP community, however, I do have strong concerns about the coarseness of the representation of the stratosphere in the model used and would appreciate the authors addressing these concerns or clearly stating the uncertainties that this may cause in their conclusions. I do appreciate the explicit representation of the bromocarbons, interactive ocean, and multiple ensembles used in this study but they still all rely on confidence in the representation of the stratosphere and its response to the forcing applied.**

We thank Reviewer_#1 for his/her support and interest on the results shown in our work, and for recognising the goodness of the explicit representation of VSL chemistry in the model. We do understand his/her concerns about the capability of CAM-Chem in representing properly the stratosphere and how it responds to the different halogen forcings. We present below a detailed point-by-point answer to each of the specific questions raised by the reviewer. We have also modified the MS accordingly, and included a CAM-Chem vs.

WACCM comparison in the Supplement. In addition to the specific answers, we accepted the reviewer´s suggestion and included the following explicit sentence in the conclusions highlighting this issue:

*"Note, however, that free-running ocean interactive simulations as the ones performed in this work possess a very large model internal variability (~10 years difference between the shortest and largest return date for run[LL+VSL]), so more ensemble members might be required to better address the important issue of the return date. Additional simulations including the explicit representation of VSL bromocarbons into Chemistry-Climate models representing the whole stratosphere would help to further reduce model uncertainties."*

**1.2   Specific Comments**

**The CAM-Chem model used in this study has 26 vertical levels and a model top around ~40km and in fig 1 state the top model level is around 5 hPa. Please add to the model description how many levels are above the tropopause. Typically models of this coarse vertical resolution have less than a dozen or so levels above the tropopause.**

We have modified the description of the model configuration (Section 2, Methods) to include the information required by the reviewer:

*"CAM-Chem was configured with a horizontal resolution of 1.9º latitude by 2.5º longitude and 26 vertical levels, from the surface up to ∼40 km (~3.5 hPa). The number of stratospheric levels changes depending on the location of the tropopause: within the tropics, there are 8 levels above the tropopause (~100 hPa), with a mean thickness of 1.25 km (15.5 hPa) for the lower stratospheric levels and 5.2 km (3.8 hPa) between the two highest levels. Within the Polar Regions, the tropopause is located approximately at ~300 hPa and up to 15 model levels belong to the stratosphere."*

**Have you done any comparisons to a model with a well resolved stratosphere like WACCM with respect to circulation, mean age, PSC area, or ClOx, BrOx, NOx, HOx concentrations? That might help to quantify uncertainties or to understand the extent that a model with so few stratospheric levels can simulate or properly represent these important quantities.**

CAM-Chem, as well as WACCM, were part of CCMVal-2 and so were included in many of the papers comparing the evolution of stratospheric ozone (Eyring et al., 2010a) as well as the model sensitivity to different greenhouse scenarios (Eyring et al., 2010b). More recently, both CAM-Chem and WACCM participated in the CMIP5 inter-comparison project, computing stratospheric ozone interactively (Eyring et al., 2013a). Note that for those studies an identical geographical and altitude configuration as the one described here was used, and CAM-Chem return dates estimations is behaving very much in the middle of the simulated return periods of the multi-model range (see Fig.1 in Eyring et al., (2010a)).

Lamarque et al. (2008) showed that even when CAM has a relatively low model top (~40 km), the model shows good ability at reproducing a variety of large- scale changes in climate and chemical composition in the stratosphere when forced with the observed sea-surface temperatures and surface concentrations of long-lived trace gases and ozone-depleting substances (more details are given in the answer to the Lamarque et al., (2012) comment below). Additionally, (Lamarque and Solomon, 2010) analysed the role of long-term increases in $CO_2$, SST and halocarbons in explaining the observed trend of ozone in the tropical lower stratosphere using CAM-Chem (v3), and compared the model performance against WACCM (see their Fig. 1, vertical distributions of the tropical vertical velocity).

Lamarque, J.-F., Kinnison, D. E., Hess, P. G. and Vitt, F. M.: Simulated lower stratospheric trends between 1970
and 2005: Identifying the role of climate and composition changes, J. Geophys. Res., 113(D12), D12301,
doi:10.1029/2007JD009277, 2008.

Lamarque, J. F. and Solomon, S.: Impact of changes in climate and halocarbons on recent lower stratosphere
ozone and temperature trends, J. Clim., 23(10), 2599–2611, doi:10.1175/2010JCLI3179.1, 2010.

CAM-Chem updates since WMO-2010 helped to improve the model performance. The
implementation of a non-orographic gravity wave (GW) scheme for convection and fronts
(originally developed for WACCM), as well as an inertia-gravity wave (IGW)
parameterization, reduced stratospheric polar temperatures (which were biased warm) and
increased chlorine activation and vortex size. As the limited vertical resolution (compared to
WACCM) does not allow the internal computation of the quasi-biennial oscillation (QBO),
the QBO is imposed by relaxing equatorial zonal winds to the observed inter-annual
variability. Additionally, stratospheric aerosol and surface area density data has been updated
to the common observation-derived dataset for the CCMI project (Eyring et al., 2013b;
Hegglin et al., 2014). A complete validation of current CAM-Chem version, focused on
tropospheric issues but including total ozone column as well as stratospheric dynamics, is
given in (Tilmes et al., 2016; see Figs. 2, 5 and 8).

We have updated the Methods section in the MS as follows:

*"The current CAM-Chem version includes a non-orographic gravity wave scheme based on*
*the inertia-gravity wave (IGW) parameterization, an internal computation of the quasi-*
*biennial oscillation (QBO) dependent on the observed inter-annual variability of equatorial*
*zonal winds, and a CCMI-based implementation of stratospheric aerosol and surface area*
*density (see Tilmes et al.(2016) for details)."*

Finally, we added in the supplement a couple of figures comparing CAM-Chem and
WACCM performance for equivalent REFC2 simulations including the additional 5 pptv
VSL[Br] contribution. The overall representation of the Total Ozone Column within the
Southern Polar Cap, as well as the Age of Air at 50 hPa validates the correct performance of
CAM-Chem in the stratosphere. We added the following lines into the MS:

*"This model configuration uses a fully-coupled Earth System Model approach, i.e. the ocean*
*and sea-ice are explicitly computed. More details of CAM-Chem performance at reproducing*
*changes in dynamics and chemical composition of the stratosphere are given in the*
*Supplementary Material."*

**Recovery of Antarctic October ozone to 1980 levels occurs in the mid 2050s in the CAM-**
**Chem simulations this is significantly earlier than the 4 models used in the WMO 2014**
**assessment which returned in the 2070s - 2080s (fig 3-15). These models had well**
**resolved stratospheres and were evaluated in CCMVal-2 to have the best representation**
**of stratospheric transport and chemistry. Why should we have confidence in the earlier**
**recovery estimate from CAM-Chem or can you appropriately caveat the conclusions**
**made with this uncertainty?**

**(2[nd] additional related comment by Reviewer_#1)**

**Page 2 lines 3-5 when discussing the Antarctic ozone return dates you reference the**
**older CCMVal-2 and WMO 2010 results and not the more recent WMO 2014 which had**
**a significantly later recovery estimate, please add mention of the WMO 2014 result here.**

We thank Reviewer_#1 for highlighting the importance of comparing our results with the last
WMO 2014 report, which present an update with respect to CCMVal-2 and WMO 2010. But we could not find any recommendation in WMO 2014 suggesting the Antarctic return date lying between 2070-2080. Indeed, the first bullet within the WMO Scientific Summary respect to Future Changes on Polar Ozone states that (WMO, 2014; Chapter 3, p3.2):

…"Arctic and Antarctic ozone abundances are predicted to increase as a result of the expected reduction of ODSs. A return to values of ozone in high latitudes similar to those of the 1980s is likely during this century, with polar ozone predicted by CCMs to recover about 20 years earlier in the Arctic (2025–2035) than in the Antarctic (2045–2060). Updated ODS lifetimes have no significant effect on these estimated return dates to 1980 values."…

Thus, there is no apparent difference on the 1980 return date recommendation between WMO 2010 and WMO 2014. Note that the estimated return dates obtained with CAM-Chem lie exactly on the (2045-2060) range given in both reports. Later, on page 3.31 and 3.32 of WMO 2014, it is made clear that the intention of Fig. 3-15 is to highlight that the SPARC 2013 updates on CFCs lifetimes do not possess a large impact on the future recovery of polar ozone. Literally:

…"Note that the differences are small and that they lie largely within the one standard deviation range, thus suggesting that the ODS lifetime change had no significant impact on the polar ozone recovery in either the Northern or Southern Hemisphere. However it should be noted that this "by chance ensemble" provides a MMM that is returning late to 1980s ozone values in the Southern Hemisphere, compared to the full WMO (2011) MMM."…

We agree with Reviewer_#1 that the 4 selected models shown in Fig. 3-15 (one of them being WACCM) are showing a delayed return date to 1980 levels for the Southern Polar Cap. Within those 4 models, WACCM (red line) return date occurs at 2060, while the 1-sigma shaded area expands all the way down to 2050. However, the 1980 baseline ozone column on Fig. 3-15 is at ~340 DU, while Fig. 2A in the MS shows a $TOZ^{SP}$ value of ~300 DU for year 1980. Evidently, the absolute return date depends on the defined ozone level prevailing at 1980, which rapidly varies between the mid-seventies and mid-nineties. Fig. S1 in the Supplementary Material show the evolution of $TOZ^{SP}$ for equivalent REFC2-CCMI simulations computed with both CAM-Chem and WACCM, which show an excellent agreement for the whole modelled period. The excellent WACCM vs. CAM-Chem comparison in the stratosphere gives confidence on the validity of the results presented in this work.

In order to explicit include the WMO 2014 recommendations in the validation of our estimated return dates, we have modified the MS as follows:

*"The multi-model CCMVal-2 ozone assessment (Eyring et al., 2010a) determined that the Antarctic ozone return date to 1980 values is expected to occur around years 2045−2060, while the impact of halogenated ozone depleting substances (ODS, such as $LL^{Cl}$ and $LL^{Br}$) on stratospheric ozone photochemistry will persist until the end of 21$^{st}$ century. Even when the 2045-2060 Antarctic return date is currently the recommended projection within the latest Ozone Assessment Reports (WMO, 2011, 2014), enhancements of stratospheric sulfuric aerosols and/or the uncertainties on greenhouse gas loadings will be especially important for stratospheric ozone recovery during the 2$^{nd}$ half of the century."*

**Page 3 lines 22-24 I couldn't find a figure in Lamarque et al. 2012 that shows reasonable overall stratospheric circulation from including the integrated momentum flux that needs to be in a model with such a low upper boundary. Can you cite or include figures that compare these simulation to observations of mean age or other measures of stratospheric circulation or transport? How is this handled in the future is it interactive or fixed. Is the circulation change over time comparable to models with a well resolved stratosphere.**

We apologise to the reviewer for citing an incorrect reference, and appreciate his/her commitment to follow the cited article to check our model validation. The correct reference, which is now properly cited in the MS is (Lamarque et al., 2008). This work was aimed at understanding the mechanisms that drive observed trends in the lower stratosphere between 1970 and 2005, based on CAM v3 model simulations.

Fig. 18b on Lamarque et al., (2008) shows the zonal mean linear trend of the January-March zonal wind tendency due to gravity wave breaking, which has the effect of increasing momentum deposition where the gravity waves break. Additionally, the latitudinal variation of the mean age of air between 100 and 3.5 hPa is also shown in Fig. 17.

The gravity wave impact on stratospheric circulation is computed interactively in the model, obtaining an overall consistent agreement with WACCM. Please, also refer to the 1[st] answer given above and to the new figures in the Supplementary Material supporting CAM-Chem performance in the stratosphere.

Lamarque, J.-F., Kinnison, D. E., Hess, P. G. and Vitt, F. M.: Simulated lower stratospheric trends between 1970 and 2005: Identifying the role of climate and composition changes, J. Geophys. Res., 113(D12), D12301, doi:10.1029/2007JD009277, 2008.

**Could you explain in the paper with a model top at around 5 hPa (from figure 1) how do you represent the 5-8% of total column ozone above the model top?**

Section 2, Methods, has been modified as follows:

*"To have a reasonable representation of the overall stratospheric circulation, the integrated momentum that would have been deposited above the model top is specified by an upper boundary condition* (Lamarque et al., 2008). *A similar procedure is applied to the altitude-dependent photolysis rate computations, which include an upper boundary condition that considers the ozone column fraction prevailing above the model top."*

**Can you show or discuss how much Br goes through the tropical tropopause in these two sets of simulations are they consistent with published aircraft and satellite estimates when VSL Br is accounted for. How well is polar BrO columns modeled compared to observations in CAM-Chem.**

Figure 1 of the original MS shows the stratospheric bromine loading due to LL and VSL sources, as well as for LL chlorine. In order to explicitly validate the halogen burden in the text, we have modified the 1[st] paragraph of the result Section 3.1 as follows:

*"The dominant anthropogenic $LL^{Cl}$ and $LL^{Br}$ scenarios included in our REFC2 simulations (Tilmes et al., 2016) show a pronounced peak at the end of the $20^{th}$ century and beginning of $21^{st}$ century, respectively, after which both their abundances decline. The respective stratospheric abundances for $LL^{Cl}$ and $LL^{Br}$ for year 2012 are approximately 3260 ppbv and 15.4 pptv, in excellent agreement with the last (WMO, 2014) report. In comparison, the evolution of $VSL^{Br}$ sources remains constant in time, with a present-day fixed contribution of ~5 pptv (Ordóñez et al., 2012). Added together, $LL^{Br} + VSL^{Br}$ show a stratospheric abundance of ~20.4 pptv at present time, in line with* Fernandez et al. (2014) *who validated CAM-Chem bromine abundances and stratospheric injection for year 2000 based on a multiple set of Specified Dynamics (SD) simulations."*

Please refer to the answer given to Reviewer_#2 (p4,l28; p11,l18) to complement our response here.

**Figure 3 Why is Spring Aug.-Oct. rather than SON**

We understand the reviewer finding out this spring definition quite un-common. The Antarctic hole formation is controlled by two different process: The chemical reactions and the physical-dynamical processes controlling the vortex formation and breakage as well as the stratospheric temperatures. As current work is mainly focused on the chemical perturbations of $VSL^{Br}$ on the ozone hole, we rather focused on the initial spring-months where the ozone hole depth is mainly controlled by the chemical component. From November on, the independent evolution of the polar vortex (which is dynamically driven) within each ensemble run is very variable and affects the ozone hole evolution very differently, with a very small dependence on the $VSL^{Br}$ loading existent at that time. Also, as the Southern Polar Cap area extends up to 63ºS, the photochemical ozone destruction begins during August, peaks during September and maximizes its overall depth in October. Thus, we used this un-common definition with the aim of highlighting the $VSL^{Br}$ contribution during the specific months when its impact is maximized. A complementary answer to this issue is given in the response to the $2^{nd}$ reviewer below.

**On Figure 5 there appears to be a significant difference in the early 1980s in ozone hole area between the observations and CAM-Chem simulations but I didn't see this mentioned in the text. Would you expect an underestimation of ozone hole area to be significant to the earlier return date found in CAM-Chem. How does this impact your conclusions?**

We thank a lot reviewer_#1 for detecting the difference in Ozone Hole Area (OHA) for the early years. We had a bug in the post-processing code that unintentionally imposed NANs (Not a defined Number) values for the date and OHA arrays before year 1990 for each of the independent simulations, which affected the ensemble mean value. We have now fixed the bug in the code and found an even better reproduction of satellite-derived OHA for the early years. Additionally, we have included in the Supplementary Material a new figure showing the OHA and OMD (Ozone Mass Deficit) validation for each of the ensemble members, including both the smoothed and non-smoothed data (see answer to large-scale oscillations below).

**Figure 6 There appears to be large 30-year time scale variability in the polar cap ozone in the ensemble average is this coming from the ocean, can you explain. The panels with the time axis show dotted lines at 2000 and 2050 but if the label is correct on the other panels you are meaning to highlight 2030 instead. Same on figure 4 and fig 10.**

We really thank the reviewer_#1 for highlighting the inconsistency between the vertical line for year 2050 and the zonal mean vertical distributions for year 2030. We have now corrected it on Figures 4, 6 and 10.

With regards to the large-scale oscillations observed for the ozone time series, they appear randomly in the smoothed fit of each of the independent simulations at different years. Even when the oscillations are reduced when the ensemble mean is computed, they still appear when the difference between $sim^{LL+VSL}$ and $sim^{LL}$ are computed (as well as when the difference between any couple of independent simulations is computed). We've tried to address this unexpected behaviour by performing different type of smoothing (moving average, hamming filter, etc.) and/or the average window considered (between 5 and 20 years) and found no dependence on the filter nor the smoothing window used. Thus, we understand these random oscillations are due of the different model variability between individual ensemble members. We also performed a power spectrum analysis to recognise the existence of a continuous wavelet oscillation on the output data, but could not assign the existence of neither a 30-year nor a 11-year signal (as suggested by Reviewer_#2).

[Figure]

Note that many other papers showing the evolution of stratospheric ozone levels (Eyring et
al., 2010a; Sinnhuber and Meul, 2015; Oman et al., 2016) show an oscillative behaviour as
the one observed for our absolute ozone trends, but none of them show any panel with the
differences between a couple of independent simulations. We wonder whether this is an issue
also existent in the output of other climate simulations performed with other type of models.

In order to make this point clear, we added a 9-pannel figure in the supplement showing the
$TOZ^{SP}$ evolution for each pair of the individual $run^{LL}$ and $run^{LL+VSL}$ simulations, including
both smoothed and non-smoothed results. We also modified the text as follows:

*"The 1960-2100 evolution of the total ozone column within the southern polar cap ($TOZ^{SP}$,*
*between $63°S-90°S$) during October is illustrated in Fig. 2. Biogenic $VSL^{Br}$ introduce a*
*continuous reduction in $TOZ^{SP}$ that exceeds the model ensemble variability between $run^{LL}$ and*
*$run^{LL+VSL}$ experiments, and improves the overall model-satellite agreement (Fig. 2a). An*
*individual panel for each independent simulation is shown in the Supplementary Material."*

*...*

*"Our CAM-Chem results show that the range in the return dates for the different ensemble*
*members of $run^{LL+VSL}$ can be of almost 10 years (i.e., of the same magnitude as the $VSL^{Br}$*
*enlargement suggested by previous studies), highlighting the importance of considering a*
*multi-member ensemble mean when performing a future return date computation. Note that*
*the return date shift for each individual simulation varies randomly independently of*
*considering or not the smoothing filter (see Figs. S2 and S3 in the supplement)."*

*...*

*"The agreement to the monthly mean ozone mass deficit (OMD) and OHA values obtained*
*from the NIWA-BS database (Bodeker et al., 2005) is largely improved when $VSL^{Br}$ are*
*considered (non-smoothed output for each independent simulation is shown in the*
*Supplementary Material)."*

**Page 3 lines 13-15 For readers unfamiliar with CCMI-REFC2 can you state the GHG**
**and ODS scenario used in this study.**

We have explicitly included in the Methods section the specific GHG and ODS scenarios as
follows:

*"At the model surface boundary, zonally averaged distributions of long-lived halocarbons*
*($LL^{Cl} = CH_3Cl, CH_3CCl_3, CCl_4, CFC-11, CFC-12, CFC-113, HCFC-22, CFC-114, CFC-115,$*
*$HCFC-141b, HCFC-142b$ and $LL^{Br} = CH_3Br, H-1301, H-1211, H-1202$ and $H-2402$) based*

*on the A1 halogen scenario from WMO, (2011) are considered, while surface concentrations of $CO_2$, $CH_4$, $H_2$, $N_2O$ are specified following the moderate Representation Concentration Pathway 6.0 (RCP6.0) scenario (see Eyring et al. (2013) for a complete description of REFC2-CCMI setup)."*

**Page 4 lines 13-15 for the total column ozone database please state which version used (is it the latest) and what years it covers. If it continues through 2015 can figures 2 and 3 be extended to include more recent years.**

We used version 2.8 of the Bodeker Scientific (NIWA) database for comparison of the Ozone Hole Area (OHA) computations. Even when there is an updated version (3.0) including data until 2015, the new version provides only unpatched daily data (without spatial or temporal interpolation). Using v3.0 would have implied to perform a "user defined" long-patch procedure, which would have made very difficult for other groups to reproduce results exactly as performed for this study. In order to compare our modelling results against the direct available data existent at present time, we decided to use the monthly mean patched data available for version 2.8 until equivalent data is available for the newest version. See comment below related to the new v3.0 database at:

http://www.bodekerscientific.com/data/total-column-ozone:

.."At this time only daily 'unpatched' data are available. We are working on generating monthly mean and patched data files as had been available in previous versions of the database. This is now a little more challenging as we intend to capitalize on the uncertainty estimates being available to calculate monthly means and patched data that incorporate realistic uncertainties. If you need the monthly mean or patched data, please continue to use version 2.8 of the database for now (see below)."…

We have modified the MS to describe the NIWA-BS database version used for comparison, as well as to include an additional comparison with non-smoothed data:

*"Model results have been compared to the National Institute for Water and Atmospheric research – Bodeker Scientific (NIWA-BS) total column ozone database (version 2.8), which combines measurements from a number of different satellite-based instruments between 1978 and 2012 (Bodeker et al., 2005)."*

*...*

*"The agreement to the monthly mean ozone mass deficit (OMD) and OHA values obtained from the NIWA-BS database (Bodeker et al., 2005) is largely improved when $VSL^{Br}$ are considered (non-smoothed output for each independent simulation is shown in the Supplementary Material)."*

**Page 8 line 19-20 4 years doesn't agree with difference 2047 and 2054 in the text. I think you meant to write 2051 instead of 2054.**

You are correct. Thanks a lot for spotting this un-intentional error.

**Given that the largest differences were found in the periphery of the ozone hole does the definition used 63-90 vs 60-90 make any difference in your dates. I have seen both regions used so either is fine, I would just suggest checking that it doesn't make a difference.**

We appreciate this suggestion on the $TOZ^{SP}$ definition. We performed the geographical integration of the total ozone column within the Southern Polar cap ($TOZ^{SP}$) for different peripheral limits, including 60ºS and 63ºS, and found no differences on the return date nor the ozone depth computed. We further performed a variable latitudinal-dependent TOZ$^{SP}$ computation, with the intention of determining the ideal outer limit definition, but no interesting results were obtained from such analysis. We then decided to use the outer limit at lat = 63ºS as other works used that definition, including the Solomon et al. (2016) healing paper to which we compare our results.

**Page 9 line 2 change deepest to deep Page 9 line 17 change "respect to" to "with respect to" Page 10 line 5 same as above.**

Thanks a lot for these corrections, which have now been included in the MS.

**************************************************************

**2    Reviewer_#2_BMS**

**************************************************************

**The study by Fernandez et al. applies the CAM-Chem chemistry climate model to**
**investigate the impact of oceanic emissions of very short-lived brominated source gases**
**(VSLS_Br) on the Antarctic ozone hole during the 21st century. This is a very thorough**
**and well performed study and the paper is well written. Its analyses help to further**
**understand results of previous studies and demonstrate the importance of oceanic**
**VSL_Br emissions for stratospheric ozone. I suggest publication in Atmos. Chem. Phys.**
**after consideration of the following comments.**

We would like to thank Bjoern-Martin for his very constructive comments.

**2.1   Specific Comments**

**For the comparison to the results of the recent study by Oman et al., Table 1 is**
**revealing, showing a difference of almost 10 years in return dates for different ensemble**
**members using the same boundary conditions. (Hope I understood this correctly.) I**
**suggest to make this point even clearer when discussing the differences to Oman et al.**

We find your appreciation very pertinent and have included a sentence highlighting this issue
both in Section 3.2 and the Conclusions:

*"Thus, the Antarctic ozone hole return date, determined following the standard computation*
*relative to the ozone column existent in October 1980 (Eyring et al., 2010a, 2010b), is not*
*significantly affected by the inclusion of natural $VSL^{Br}$ sources. This result contradicts the*
*recent findings from Yang et al. (2014) and Oman et al. (2016), who estimated an increase*
*between 7 to 10 years on the ozone hole return date. Note, however, that the former study*
*performed non-coupled (without an interactive ocean) timeslice simulations including a*
*speculative doubling of $VSL^{Br}$ sources on top of background $LL^{Cl}$ and $LL^{Br}$ levels*
*representative of years 2000 and 2050, while Oman et al. (2016) considered a single member*
*climatic simulation for each type of experiment and thus lacks an assessment of the internal*
*model variability. Our CAM-Chem results show that the range in the return dates for the*
*different ensemble members of $run^{LL+VSL}$ can be of almost 10 years (i.e., of the same*
*magnitude as the $VSL^{Br}$ enlargement suggested by previous studies), highlighting the*
*importance of considering a multi-member ensemble mean when performing a future return*
*date computation. Note that the return date shift for each individual simulation varies*
*randomly independently of considering or not the smoothing filter (see Figs. S2 and S3 in the*
*supplement)."*

*...*

*"Note, however, that free-running ocean interactive simulations as the ones performed in this*
*work possess a very large model internal variability (~10 years difference between the*
*shortest and largest returned date for $run^{LL+VSL}$), so more ensemble members might be*
*required to better address the important issue of the return date."*

**The effect of VSLS_Br maximizes in the late 1990s (e.g., Figs 4c and 6c), but there is a**
**secondary maximum around 2030 (and following minor maxima around 2060 and 2090).**
**Why is that? Is this an artifact from the 11-year smoothing?**

Reviewer_#1 also noticed this 30-years oscillation on the ozone differences. Please refer to
the answer given above.

**p2,l20: Reference to Sinnhuber and Meul might be slightly misleading: They showed, that indeed the highest impact is during periods of high aerosol loading, but the strongest impact on ozone depletion is not at mid-latitudes, but at the Antarctic ozone hole.**

True, and in order to avoid misleading interpretations, we have replaced the text as follows:

*"The additional stratospheric contribution of biogenic VSL$^{Br}$ improves the model/observations agreement with respect to stratospheric ozone trends between 1980 and present time (Sinnhuber et al., 2009), with* large *ozone depleting impacts during periods of high aerosol loading within mid-latitudes (Feng et al., 2007; Sinnhuber and Meul, 2015)."*

*...*

*"More recently, Sinnhuber and Meul, (2015) found that the impact of VSL$^{Br}$ maximize in the Antarctic Ozone hole (~20% greater ozone depletion), while Oman et al., (2016) determined that the addition of 5 pptv VSL$^{Br}$ to the stratosphere could delay the ozone return date to 1980 levels by as much as one decade."*

**p4,l28: "…the increase in SST and atmospheric temperature … is expected to … additionally enhance the stratospheric injection of VSL_Br": This effect should already be included in the current simulations, so would not be additional, as I understand?**

**p11,l18: "… or even more if the oceanic VSL_Br source strength and deep convection increases …": For the deep convection, I assume this is already considered here (see my comment above), while it should be acknowledged that the increase in oceanic source strength is largely speculative at this point.**

As current work is focused on Antarctic Ozone, our original draft does not include an in-depth analysis of the evolution of VSL species on the tropical regions where most of the stratospheric injection occurs. Both reviewers have simultaneously addressed the importance of understanding the extent at which this "additional" VSL enhancement through changes in deep convection is occurring, something that we are describing in detail in another forthcoming paper. As the additional impact of VSL$^{Br}$ on Antarctic Ozone depends on the total amount of biogenic bromine injected, we prefer to avoid discerning between source gas (SG$^{VSL}$) and product gas (PG$^{VSL}$) partitioning in this work, because a complete treatment of stratospheric injection must include additional factors (SST, emissions variability, age-of-air, convection, etc.). The additional enhancement of VSL$^{Br}$ stratospheric injection, as we conceive, must include a detailed analysis of the inorganic fraction of VSL bromine (PG$^{VSL}$) being injected.

Preliminar results indicate that even when there is a gradual change in the bromine partitioning between carbon-bonded (SG$^{VSL}$) and inorganic (PG$^{VSL}$) species as we move into the 21$^{st}$ century, the total bromine injection of VSL$^{Br}$ occurring at the tropical tropopause remain practically constant with time. Thus, the faster transport of air masses from the ocean surface to the tropical tropopause layer, seems to reduce the photo-degradation of the dominant VSL$^{Br}$ organic sources, increasing the less reactive carbon-bonded fraction. To make this issue clear in the text, we have removed the term additionally and modified it as follows.

*"Knowledge of the extent at which the inorganic fraction of VSL$^{Br}$ is being injected to the stratosphere is of great importance as it strongly affect the ozone levels mostly in the lowermost stratosphere (Salawitch et al., 2005; Fernandez et al., 2014), which has implications at the altitudes where the strongest O$_3$-mediated radiative forcing changes due to greenhouse gases are expected to occur (Bekki et al., 2013). Note that the atmospheric*

*burden of the inorganic bromine portion in the tropical tropopause layer is highly dependent on the competition between heterogeneous recycling reactions, evaporation and washout processes occurring on the surface of ice-crystals (Aschmann et al., 2011; Fernandez et al., 2014)."*

As for the speculative future evolution of VSL emissions, we have included it in the Methods sections when describing the scenarios used for the study.

*"In order to avoid unnecessary uncertainties associated to the speculative evolution of VSL$^{Br}$ oceanic emissions, we used a constant annual source strength for the whole modelled period."*

**p5, ozone hole evolution: Do the model simulations include volcanic eruptions or not? Would be good to mention during the discussion of Fig.2, as Pinatubo may have played a role.**

Our REFC2 simulations follow the CCMI guidelines described in detail in Eyring et al., (2013), thus they include implicit representation of volcanic eruptions in the past but not for the future. We find not necessary to distract the attention into the specific impact of Pinatubo eruption on the Antarctic Ozone Hole, but we will do on a forthcoming work on the impact of VSL on the global stratosphere. To avoid any misinterpretation on this topic, we have modified the text within the Methods section as follows:

*"Note that our REFC2 setup includes volcanic eruptions in the past, but possible volcanic eruptions in the future are not considered, as they cannot be known in advance (Eyring et al., 2013b)."*

**Fig. 3: The separation into different seasons is very helpful, but why is spring defined as AUG-SEP-OCT, instead of SEP-OCT-NOV, and why not include winter (JUN-JULAUG) for completeness?**

In order to highlight the seasonal impact of VSL$^{Br}$ on Antarctic ozone we focused on those months where the chemical component of ozone destruction dominates respect the dynamical component controlling the vortex formation/breakage (please, refer also to the answer given to Reviewer_#1 above). Thus, we decided to compute the seasonal average considering those months where the chemical impact is not strongly affected by the physical changes produced by a different dynamical evolution of the polar vortex within each ensemble run. In this way, we did not include August during Winter, as the CAM-Chem monthly output includes the initial springtime ozone depletion occurring at low latitudes (the polar cap definition extends up to 63ºS). Similarly, November and December are not considered in Spring and Summer, respectively, as during those months usually occurs the vortex breakage. Even when for the Fall there are not any dynamical factor of interest to consider, we decided to compute a bi-monthly average in concordance with the remaining panels. We accepted the suggestion and added the JUN-JUL panel for completeness, although the Bodeker database has NANs values for the Southern Polar Cap region during those months.

In order to make these points clear, we have modified the MS and figure caption as follows:

*…"Agreement between model and observations for TOZ$^{SP}$ and $\Delta TOZ^{SP}_{1980}$ improves for all seasons when VSL$^{Br}$ are considered (Fig. 3). To highlight the additional chemical destruction of Antarctic ozone due to biogenic bromine, the monthly output where for those months where ozone depletion is dynamically controlled by the polar vortex formation and breakage (i.e., August and November/December, respectively) had been discarded."…*

...

*"Figure 3: Idem to Fig. 2, but computing the average for A,E) Spring (defined as SEP-OCT); B,F) Summer (JAN-FEB); C,G) Fall (MAR-APR); and D,H) Winter (JUN-JUL). The monthly output for the periods where a strong dynamical transition between seasons exists has not been considered (see text for details)".*

**2.2   Technical Corrections**

**Sometimes reference is to Carpenter et al., 2014, sometimes to WMO, 2014 (e.g., p2,l10) with no obvious reason for the distinction.**

Chapter 1 in (WMO, 2014) summarizes the current Updates on Ozone-Depleting Substances (ODSs) and Other Gases of Interest to the Montreal Protocol (Carpenter et al., 2014). In the original MS, whenever we referred to this chapter, we pointed out to (Carpenter et al., 2014), while when we were pointing at the ozone impact of VSL chemistry and/or the future evolution of the ozone layer under different emission scenarios, we cited the whole report (WMO, 2014). As we would need to cite more than 3 chapters from the report if we were to make the same distinction as for Chapter 1, we accepted the reviewer suggestion and we now only cite the whole (WMO, 2014) report at all times.

**p2,l13: Saiz-lopez -> Saiz-Lopez**

**p3,l31: "on 1950" -> "in 1950"**

**p7,l15: "at the lowermost" -> "in the lowermost" (?)**

All three corrections have been included in the revised MS.